



# Improved Monitoring of Subglacial Lake Activity in Greenland.

Louise Sandberg Sørensen[1,*], Rasmus Bahbah[1,*], Sebastian B. Simonsen[1], Natalia Havelund Andersen[1],
Jade Bowling[2,3], Noel Gourmelen[4,5], Alex Horton[5], Nanna B. Karlsson[6], Amber Leeson[2],
Jennifer Maddalena[2], Malcolm McMillan[2], Anne M. Solgaard[6], and Birgit Wessel[7]

[1]Geodesy and Earth Observation, DTU Space, Technical University of Denmark,Kgs. Lyngby, DENMARK
[2]UK Centre for Polar Observation & Modelling, Centre of Excellence in Environmental Data Science, Lancaster University, Lancaster, UNITED KINGDOM
[3]Lancaster Environment Centre, Lancaster University, Lancaster, UNITED KINGDOM
[4]School of GeoSciences, University of Edinburgh, Edinburgh, UNITED KINGDOM
[5]EarthWave Ltd., Edinburgh, UNITED KINGDOM
[6]Geological Survey of Denmark and Greenland, Copenhagen, DENMARK
[7]German Remote Data Center, German Aerospace Center (DLR), Oberpfaffenhofen, GERMANY
[*]These authors contributed equally to this work.
**Correspondence:** Louise Sandberg Sørensen (slss@space.dtu.dk)

**Abstract.** Subglacial lakes form beneath ice sheets and ice caps if water is available, and if bedrock and surface topography are able to retain the water. On a regional scale, the lakes modulate the timing and rate of freshwater flow through the subglacial system to the ocean by acting as reservoirs. More than one hundred hydrologically active subglacial lakes, that drain and recharge periodically, have been documented under the Antarctic ice sheet, while only a handful of active lakes have been

identified in Greenland. The small size of the Greenlandic subglacial lakes puts additional demands on mapping capabilities aiming to resolve the evolving surface topography in sufficient detail to record their temporal behavior. Here, we explore the potential for combining data from CryoSat-2, TanDEM-X, and ArcticDEM to document the evolution of four active subglacial lake sites in Greenland. The inclusion of the new data sources provides important information on lake activity, documenting that the ice surface collapse basin on Flade Isblink ice cap was 50% (30 meters) deeper than previously recorded. We also present

evidence of a new active subglacial lake in Southwest Greenland, which shows signs of being hydrologically connected to another subglacial lake in that region. This is to our knowledge the first evidence of hydrologically connected subglacial lakes in Greenland, indicating that water is transferred from one lake to another following a draining event. These findings show how improving the measurement capabilities of subglacial lakes, improves our current understanding and knowledge of the subglacial water system and its connection to surface hydrology.

## 1  Introduction

A subglacial lake is a body of water stored beneath an ice sheet, glacier, or ice cap. Subglacial lakes are part of the basal hydrology and drainage system and may act as buffers between the melt generated on and below the ice, and the flux to the ocean (Fricker et al., 2007; Siegert et al., 2016). The origin of the water contained by a subglacial lake depends on its regional



setting. The water that feeds subglacial lakes may be generated by ice melting caused by geothermal heat or by frictional heat
from ice flow, or surface water channeled to the bed as is the case in some regions.

The number of observed subglacial lakes is growing (Livingstone et al., 2013) and while their total volume is not large, mapping their dynamics is important to better understand the movement of meltwater through the subglacial system. Currently, 64 subglacial lakes have been discovered in Greenland, while 675 have been detected in Antarctica (Livingstone et al., 2022). The Greenland Ice Sheet (GrIS) is warmer, thinner, and generally characterized by a larger surface slope than the Antarctic
Ice Sheet (AIS), and it is possible that past subglacial lakes in Greenland drained at the end of the last glacial period (Pattyn, 2008). Additionally, subglacial lakes in Greenland are typically small and located close to the margin of the ice sheet (Bowling et al., 2019), where the rapidly evolving surface mass balance further hampers the detection of subglacial lake activity.

In accordance with Livingstone et al. (2022), we define a subglacial lake as stable if its volume remains relatively constant over time, or as being active if it is observed to periodically drain and refill. Triggering of subglacial lake drainage events can
e.g. occur after a prolonged addition of surface meltwater (Livingstone et al., 2022). The lake will eventually drain when filled with enough water to resist the pressure exerted by the overlying glacial load (Chandler et al., 2013). The sudden drainage and outburst flood of a subglacial lake might temporarily affect ice flow velocities downstream from the lake location (Palmer et al., 2015; Liang et al., 2022). Therefore, the behavior of subglacial lakes is important to consider when discussing the response of the ice sheets to a warming climate (Willis et al., 2015). The behavior of active subglacial lakes is also an important indi-
cator of hidden subglacial processes. Notably, by monitoring after a lake drainage event, the period of lake recharge provides information about subglacial water production and conditions at the bed (Malczyk et al., 2020).

Evans and Smith (1970) were the first to detect a subglacial lake under the AIS by Radio-Echo-Sounding (RES). RES can penetrate the ice sheet and map the bedrock topography, where a strong, flat reflection indicates basal water presence (Bingham and Siegert, 2007; Tulaczyk and Foley, 2020). RES has been used to detect and map numerous subglacial lakes under the ice
sheets (Wright and Siegert, 2012; Siegert et al., 2016; Bowling et al., 2019). Stable subglacial lakes cannot be identified from the characteristics of the overlying ice surface, except for large lakes which may influence surface topography as seen at Lake Vostok, AIS where the surface is exceptionally flat (Ridley et al., 1993). Active subglacial lakes, may on the other hand be identified by ice surface collapse basins (surface depressions) created when the lake drains, and localised surface uplift as the lake refills. The surface expressions of subglacial lake volume oscillations are controlled by viscous ice flow (Stubblefield
et al., 2021). Only four of the known subglacial lakes in Greenland have been identified by ice surface collapse basins, while the rest are identified by the use of RES (Bowling et al., 2019).

The small size of the subglacial lakes in Greenland ($<1$ km$^2$) makes them impossible to map from conventionally processed radar satellite altimetry (Meloni et al., 2020), whereas their large Antarctic counterparts (10-100 km$^2$) have been monitored extensively (Livingstone et al., 2022). However, due to the novel Interferometric Synthetic Aperture Radar (SARIn) mode of
the European Space Agency's (ESA) first Earth explorer mission CryoSat-2 (CS2) and recent advances in so-called swath-processing (Gray et al., 2013; Foresta et al., 2016; Gourmelen et al., 2018a; Andersen et al., 2021), we now can look at even smaller targets as suggested by Wingham et al. (2006). The swath processing enables us to generate a swath of elevation estimates across track, which increases the spatial data resolution and coverage. This processing method means that the ability





to map topographic lows is improved compared to conventional retracking which preferably tracks the point of closest approach
(topographic highs). This increases the chance of acquiring data over small surface features. An additional source of high-resolution ice surface topographic information is provided by two sources of Digital Elevation Models (DEMs); TanDEM-X derived from the X-band TanDEM-X satellite mission (Rizzoli et al., 2017), and ArcticDEM from the panchromatic band WorldView satellite mission (Porter et al., 2018). Here, we investigate the capabilities and added value of the CS2 swath-processed altimetry data and high-resolution TanDEM-X DEMs, focusing on the four active subglacial lakes in Greenland that have previously been identified in the literature (Palmer et al., 2013, 2015; Howat et al., 2015; Bowling et al., 2019). These four subglacial lakes are all characterized by the occurrence of collapse basins in the ice sheet/ice cap surface topography after a lake drainage event. Through analysis of swath-processed CS2 data, TanDEM-X DEM scenes, and ArcticDEMs we present time series of collapse basin depths and volumes at an unprecedented temporal resolution, thus advancing our understanding of the subglacial lake draining and refilling timing and rates of the four active subglacial lakes.

## 2   Subglacial Lake Sites

The four active subglacial lakes in Greenland identified by Bowling et al. (2019) from observations of surface collapse basins are located as following; one in West Greenland, two in Southwest Greenland, and one under the Flade Isblink ice cap in Northeast Greenland. Here, we summarize the present knowledge of the lakes following Palmer et al. (2015); Willis et al. (2015); Bowling et al. (2019).

### 2.1   Lake 1: West Greenland

Lake 1 is a subglacial lake located in the western part of the GrIS (67.611°N, 48.709°W) northwest of the Inuppaat Quuat glacier (Fig. 2a), and just below the Equilibrium Line Altitude (ELA). The temporal evolution of Lake 1 has previously been studied by Palmer et al. (2015) and Howat et al. (2015), using a variety of elevation data sets such as Search-space Minimization (SETSM) DEMs derived from WorldView data, the Greenland Mapping Project (GIMP) DEM, ICESat along-track elevations and airborne LiDAR data, and optical imagery. The datasets constrained the spatio-temporal evolution of the lake drainage and the associated ice surface collapse. These studies found that Lake 1 drained at an unknown rate within two weeks (28 June, 2011 to 12 July, 2011), resulting in the formation of a collapse basin in the ice sheet surface. A SETSM DEM from October 28th, 2011, revealed a collapse basin of about 1 km in width and 60-70 m in depth. The bottom of the collapse basin was flat, which suggested that the subglacial lake was still partially filled. According to both Palmer et al. (2015) and Howat et al. (2015) it is likely that Lake 1 receives meltwater from the surface, and that the drainage of Lake 1 in 2011 may have been triggered by the drainage of a nearby supraglacial lake. The routing of water to the bedrock from the surface, e.g. through a moulin, has been known to trigger drainage of subglacial lakes due to overfilling by meltwater (Willis et al., 2015). Howat et al. (2015) also found indications of a 2004 subsidence event above the lake. The collapse basin was observed to partially refill between 2011 and 2013, however, it could not be concluded whether the subglacial lake recharged or if the depression simply filled up with surface water or snow.



## 2.2 Lake 2 and 3: Southwest Greenland

Bowling et al. (2019) identified two collapse basins in Southwest Greenland, located between the Sermeq and Sioqqap glaciers (Fig. 3a and 4a). They classified with very high confidence that these surface depressions are associated with the drainage of subglacial lakes. We denote the northernmost lake as Lake 2 (63.542°N, 48.449°W), and the southern one as Lake 3 (63.261°N,

48.207°W). The two lakes are located about 35 km apart, and the collapse basin over Lake 2 was 15 m deep in August 2012, while the one over Lake 3 was 18 m deep in June 2012. Using ArcticDEM strips from 2015, they also found that both collapse basins decreased in volume in the period 2012-2015, which suggests a refilling of the subglacial lakes, and it was further estimated that the recharge of Lake 3 has been ongoing since 2001, while the drainage event for Lake 2 was not identified. Optical images show supraglacial lake drainage in the region, which could indicate that the subglacial lakes to some extent are

filled by surface water.

## 2.3 Lake 4: Flade Isblink ice cap

The collapse basin above a subglacial lake on the southern dome of Flade Isblink ice cap in the northern part of Greenland (Fig. 5a) has been described by Willis et al. (2015); Liang et al. (2022), and we denote it Lake 4 (81.157°N, 16.613°W). Willis et al. (2015) base their analysis on DEMs from stereo satellite imagery together with airborne LiDAR observations, while Liang

et al. (2022) investigate the subglacial lake using ArcticDEMs and ICESat-2 altimetry data between 2012 and 2021. From MODIS optical imagery Willis et al. (2015) find that the ice surface above Lake 4 had subsided in the autumn of 2011, leaving a surface depression shaped like a mitten. The basin was estimated to have formed over a three-week period between August 16, 2011, and September 6, 2011, and it comprises two sub-basins. The first estimate of elevation measurements was from a WorldView-1 derived DEM from May 2012 showing a maximum depth of the collapse basin of about 70 m. The elevation of

the collapse basin rapidly increased by 30 meters over the following two years due to inflow of surface water, and between August 2012 and April 2013 a topographic bulge appeared in the basin (Willis et al., 2015). Liang et al. (2022) find that the lake drained again in 2019, but that it released much less water than during the 2011 event.

## 3 Data Processing

In the following, we outline our processing steps for the remote-sensing data.

### 3.1 CryoSat-2 Data

CS2 measures in three different operational modes; Low Resolution Mode (LRM), Synthetic Aperture Radar (SAR) and Interferometric SAR (SARIn), of which the latter can be used for swath processing (Wingham et al., 2006). Here, we swath process CS2 SARIn level 1b baseline D data (Meloni et al., 2020). The conventional SARIn Level 2 (L2) elevation product from CS2 consists of a single surface elevation measurement at the Point-Of-Closest Approach (POCA) along the satellite

flight line. This L2 elevation product exploits just a fraction of the 1024 measurements contained in a single CS2 waveform.



By implementing swath processing of radar altimetry, it is possible to generate additional elevation measurements using part or all of the remaining echo (Gourmelen et al., 2018b; Andersen et al., 2021). The CS2 pulses scatter from Earth's surface from distinct locations in the satellite across track direction, and these scattering locations are recorded within the Level 1b waveforms. The principle of the L2 swath processing algorithm is to identify the high coherence data points that scatter off the

ice and to extract the corresponding elevation and location. The main steps of L2 swath processing for extracting elevations are: (1) Identifying high-quality records (e.g. based on selected coherence and power thresholds), (2) Unwrapping (removing phase jumps from the data), and (3) computing elevation and geographic location (Gourmelen et al., 2018b; Andersen et al., 2021). This leads to the generation of elevation measurements at ranges beyond the POCA location, and to an overall increase in the spatial density and coverage compared to the L2 product. This increase is needed in order to map the small-scale features

of the ice surface collapse basins above active subglacial lakes in Greenland (Noël et al., 2015; Andersen et al., 2021). To increase the chance of detecting the small-scale signatures of subglacial lake activity in Greenland, the swath processor was tuned to allow for the inclusion of elevation estimates associated with lower coherence and power thresholds than what is usually applied in the literature (Gourmelen et al., 2018b; Andersen et al., 2021). Decreasing the normalised coherence limit from 0.8 to 0.6, Andersen et al. (2021) found that the number of generated data points increased by 25%, but the standard

deviation of intra-mission crossover elevation difference increased 35-65 %.

Lowering the coherence and power increases the probability of phase unwrapping errors in the L2 elevation product (Gourmelen et al., 2018a), and filtering of the generated elevation point data is therefore required to remove erroneous data from the subsequent analysis. Here, filtering based on coherence, power and range bin numbers of each produced elevation estimate is applied. We find that the swath-processed CS2 data from within the collapse basin and from the surface near the

rim, respectively, have distinct compositions of these three waveform parameters. This allows us to identify and remove noisy data points by choosing appropriate threshold values for these parameters. The thresholds were determined by comparing CS2 swath elevation estimates derived using different threshold values, with a temporally close ArcticDEM scene that contained the collapse basin. The optimal normalised coherence and power threshold values were found to be 0.6 and 0.001, respectively.

Even after the removal of data based on this filtering, some erroneous data points are still found over the collapse basins. This

is believed to be caused by the highly dynamic surface at the collapse basins, which changes the scattering mechanisms over time, resulting in a larger incoherent component in the swath processing. This could be negated by increasing the thresholds, but it was not possible to find an ideal combination that removed every error and also keeping the data.

These apparent errors are therefore removed in a second step of the filtering by applying a lowest-level filtering to the elevation estimates within the outlined collapse basin, so that the cluster of data with the lowest elevation within the defined

lake outline are assumed to representative of the bottom of the collapse basin. The removed estimates were deemed as errors after we compared them with ArcticDEMs that were close in time to the CryoSat-2 SARIn data. Across swath tracks close to the basin rim and the slanting walls would often give erroneous estimates compared to the ArcticDEMs and were removed. We find that the fraction of data, which is removed, depends on the scattering mechanisms of the basin and the distance to the CS2 nadir track.





## 3.2 TanDEM-X Data

TanDEM-X is an interferometric synthetic aperture radar (InSAR) system consisting of two satellites, TerraSAR-X and TanDEM-X, launched in 2007 and 2010, respectively. Its primary mission objective was the generation of a global digital elevation model, which was completed in 2016 with a spatial resolution of 0.4 arc seconds (i.e., about 10 m - 12 m) (Rizzoli et al., 2017). Here, we use TanDEM-X data acquired between the years 2010 and 2017. The InSAR data were requested via a TanDEM-X science proposal as co-registered single-look complex data, and the interferometric processing and calibration to produce interferometric DEM scenes was done by the German Aerospace Center (DLR) using the Integrated TanDEM-X Processor (ITP) and the Mosaicking and Calibration Processor (MCP) (Rizzoli et al., 2017). The ITP processed the interferometric bistatic data to interferograms and then performed phase unwrapping and geocoding (Lachaise et al., 2018; Rossi et al., 2012). The absolute vertical calibration of the uncalibrated DEM scenes was performed by MCP block adjustment. For Greenland, this procedure relied on ICESat points over rocks as vertical reference and tie points to transfer the height level to the data takes located further inland (Wessel et al., 2016). For each DEM scene, an individual height offset was estimated and applied (up to 10 m). The penetration of the X-band SAR signal into the snow and ice surface by several meters (Rott et al., 2021; Fischer et al., 2020; Wessel et al., 2016) was maintained by the block adjustment and may therefore complicate validation and comparison with other data. However, this issue was by-passed in this study by aligning different DEMs at stable anchor points, which is described in Section 4.1.

## 3.3 ArcticDEM

The ArcticDEM input data comprise of timestamped 2 m spatial resolution DEMs covering the period 2011-2017 (Porter et al., 2018). These DEMs have been generated from stereoscopic WorldView and GeoEye satellite imagery by the ArcticDEM Team, using the Surface Extraction with TIN-based SETSM algorithm (Noh and Howat, 2018). Following their generation, they are freely distributed as strip files to the community by the US Polar Geospatial Centre. The DEMs are then co-registered using lateral and vertical corrections provided within the ArcticDEM metadata (Porter et al., 2018).

## 4 Methods

### 4.1 Alignment of data sets

We vertically align the three elevation data sets (TanDEM-X, CS2, and ArcticDEM) to account for fluctuations in the surface elevation caused by regional surface mass balance (SMB), ice dynamics (similar to what was done in (Palmer et al., 2015)) and for different penetration biases for the radar measurements, typically 0.5 m - 1 m for CS2 and 3 m - 4 m for TanDEM-X (Wessel et al., 2016; Abdullahi et al., 2019). The vertical alignment is done by defining anchor points close to the rim of each subglacial lake surface depression in the earliest available DEM. All consecutive DEMs are height corrected to align with the reference DEM at the location of the chosen anchor point. To vertically align the discrete CryoSat-2 swath data to the DEMs, the median of the difference between CS2 points close to the collapse basin and the reference DEM were used to correct for





the vertical the bias. This procedure ignores different penetration depths at the surface on the rim, but it preserves the relative heights from the top of the rim to the bottom of the depression for each sensor. This alignment allows us to focus on the local height change within the collapse basin likely caused by the dynamic hydrological processes. The maximum vertical alignment needed was found to be 12 m, but for the vast majority of the data sets the alignment correction was less than 3 m.

The data are also aligned horizontally to account for ice flow. Lakes 1-3 are located in the upper ablation zone in the southwestern part of the GrIS, where the ice flows in a westerly direction. Optical imagery as well as the three data sets provide evidence that the collapse basins advect with the ice flow. Figure 1 shows the drift of the collapse basin over Lake 1 in the period 2011-2021. ArcticDEMs were used to create the collapse basin outlines between 2011 and 2017, while Landsat-8 was used until 2021. We compared an ArcticDEM scene and a Landsat-8 image from the summer period of 2015, and found no

considerable visual bias in the basin outline. The basin drifts in a westerly direction, and decreases 95 % in surface area from 2011-2021 as the subglacial lake gradually recharges, leading to the collapse basin moving away from the subglacial lake location. This movement has created a surface depression about 100 meters downstream in 2021, ten years after the drainage.

To consistently track the evolution of the depth of the collapse basins, even when they change location due to ice flow, we horizontally align the data sets by correcting for the observed local ice flow using MEaSUREs Greenland Ice Sheet Velocity

Map from InSAR Data (Joughin et al., 2010, 2015) and Greenland Ice Velocity from Sentinel-1 data, Edition 2 (Solgaard et al., 2021; Solgaard and Kusk, 2021). At the location of Lake 4, the ice flow velocity is found to be <17 m/yr but we see no evidence in the elevation that this collapse basin has moved over time. One reason for this can be that the subglacial lake drains again in the observational period, and also the larger size of this lake makes the potential ice flow less evident. No horizontal alignment was made at Lake 4.

## 4.2   Time Series of Deepest Point

The temporal evolution of the depth and shape of the collapse basins is controlled by local factors such as refilling of the subglacial lake, SMB, and ice dynamics. Here, we determine the location of the deepest point within a basin. Lake 4 is the largest lake with a relatively flat 1000 m bottom diameter in contrast to the 100 m to 400 m ground diameter of Lake 1-3. We define the deepest point based on the first DEM in which the basin is detected, and use the temporal change in elevation of this

location as a measure for the evolution of the subglacial lake. For each time-stamped ArcticDEM and TanDEM-X, all points within a distance of 50 meters from the initially deepest point were sampled as this ensures enough data to calculate a robust mean and standard deviation ($\sigma$), while still avoiding sampling of the slanting basin walls. Due to the scattered coverage and the limited number of CS2 data points, the corresponding estimate of the basin depth at Lake 1 is derived from all CS2 point data within the outline of the basin. We only used CS2 track crossings from which 10 or more points are available within

the basin, and points within $5\sigma$ of all data within the basin. The horizontal alignment and the filtering of the CS2 swath data, ensures that we do not need to change the basin shapefile through time. At the substantially larger Lake 4, we sampled all CS2 points within 400 meters from the deepest point, thus not sampling the inclining basin floor. We compute the mean of each of the sampled data sets and use this as the value for the elevation of the deepest point. Their standard deviation represents the




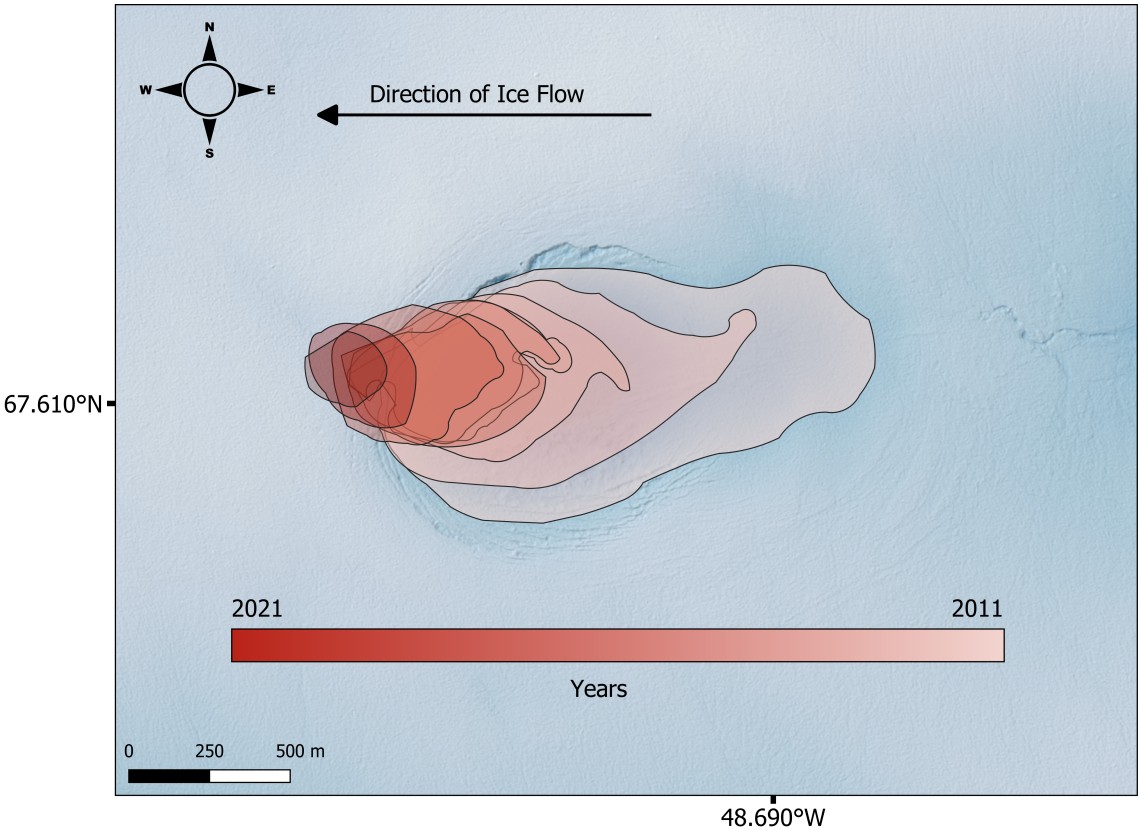

**Figure 1.** The horizontal drift of the collapse basin over Lake 1 during the period from 2011 to 2021. Its location is shown in Fig. 2. ArcticDEMs were used to create the collapse basin outline between 2011 and 2017, while Landsat-8 images were used until 2021. The basin drifts in a westerly direction and the surface area decreases substantially.

spatial variability of these data, and we use $2\sigma$ as the error bar on the depth estimate, but note that this is not a measure of their
true accuracy.

### 4.3 Subglacial Lake Volumes

We estimate subglacial lake volumes by assuming that the subglacial lake volume changes are directly transferred to the surface topography. While this is an approximation, the study by Stubblefield et al. (2021) supports this, demonstrating that at an ice thickness of ~1 km (500 m - 1.2 km at our study sites) and draining time of less than half a year (we find it to be within weeks
at study sites), the volume of the lake and the collapse basin will be almost the same. We derive the collapse basin volume by extracting surface elevations of the basin area using the TanDEM-X and ArcticDEM DEMs manually delineate the collapse basin outline from the earliest available DEM, which contains the collapse basin. The extracted DEM height anomalies are derived by subtracting the median height at the basin rim as defined by the manually delineated outline. The volume of each



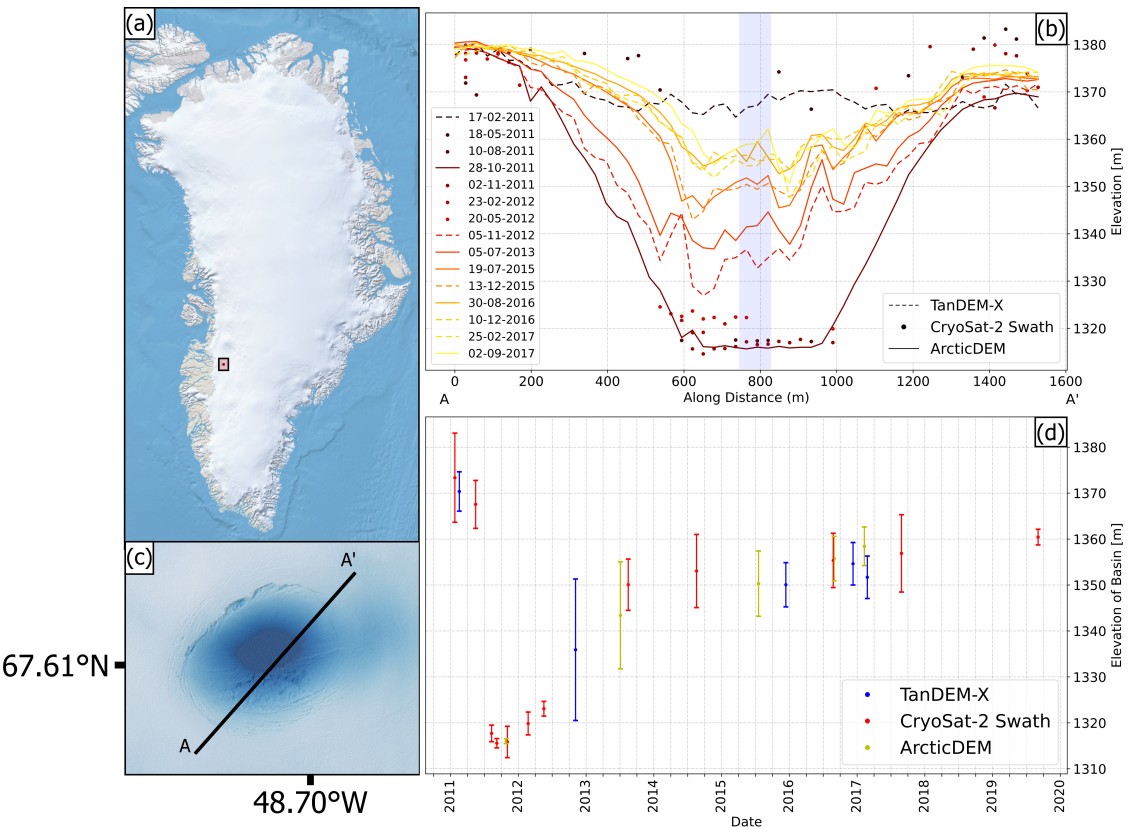

**Figure 2. Lake 1 :** (**a**) The location of the collapse basin. (**b**) Elevation profiles from the three aligned data sets, TanDEM-X is the dashed line, CryoSat-2 data is represented by dots and ArcticDEM is the solid line. **c**) The collapse basin as seen in an ArcticDEM, 28th of October 2011. Line A-A' is the profile used in Figure (b). (**d**) Time series of the deepest point of the lake basin from TanDEM-X (blue), CS2 (red) and ArcticDEM (yellow).

grid point within the outline is summed to provide a total basin volume. The horizontal and vertical DEM alignment (described

in Sect. 4.1) ensures that the original outline also can be used to derive the temporal changes in basin volumes. The uncertainty of the DEM volumes are estimated by cubing the $2\sigma$ from the depth estimates.

At collapse basins where CryoSat-2 swath data is also available, we use a different approach: Based on each available DEM over a given lake, we calculate the shape factor ($R$) between lake depth ($H$) and volume ($V$) at times $t_i$:

$$R(t_i) = \frac{V(t_i)}{H(t_i)} \tag{1}$$

Since the collapse basin shape and form changes over time, the shape factor $R$ changes over time, and we fit a smoothed function, $\widetilde{R}(t)$, through all available $R(t_i)$ values, taking their error bars into account. We the use $\widetilde{R}(t)$ to estimate lake volumes from the CS2 depth estimates ($H_{CS2}(t)$) when available, through:



$$V_{CS2}(t) = \widetilde{R}(t) \cdot H_{CS2}(t), \qquad (2)$$

to construct a time series of lake volumes constrained by both the available DEMs and CS2 data with their depth uncertainties. To account for the CS2 uncertainties in the volume estimation, the uncertainty at all $V_{CS2}(t)$ were computed by using the depths $H_{CS2}(t) \pm 2\sigma$, where $\sigma$ is the standard deviation of the CS2 depth mean.

## 5 Results

For each lake site, we present the temporal evolution of the ice surface collapse basins both by height profiles across the basins and as a time series of the elevation of the deepest point in each basin. Estimated lake volume changes are presented at the end of this section.

### 5.1 Subglacial Lake Depths

Our findings for Lake 1 are presented in Figure 2. Figure 2b shows the ice sheet elevations of Lake 1 along the A-A' profile shown in Figure 2c. The location of A-A' is adjusted through time to account for ice flow. With CS2 swath processing, it was possible to extract point measurements from 12 satellite tracks passing over Lake 1. The CS2 data are point observations of elevations and do not provide a full surface coverage as the TanDEM-X and ArcticDEM DEMs do. Hence, the CS2 observations plotted are those data points that are located closer than 180 m to the A-A' profile, as this distance ensures that data from all satellite crossings are represented. The colours of the line indicate the time of the measurements, with darker colours for the start of the measurement period (2011) and lighter colours for the end of the measurement period (2017). We find that a TanDEM-X DEM from February 17, 2011, and CS2 data from May 18, 2011, show that that the surface is generally flat, indicating that the ice surface collapse basin has not yet been created. The first data set that shows signs of a collapse basin is a CS2 crossing from August 10th, 2011, which detects a depression of the surface of approximately 60 m, and which has a flat surface in the bottom. Over time, the collapse basin decreases in size (both in depth and area), but a depression is still evident in the most recent displayed data set, which is an ArcticDEM from September 2nd, 2017. For clarity, we have not shown all available data sets in Figure 2b. Figure 2d shows the temporal evolution of the depth of the deepest part of the collapse basin (Sect. 4.2), and it shows that the basin depth did remain stable during the winter of 2011/2012, following the collapse. In the period from February 2012 to July 2013 TanDEM-X and ArcticDEM DEMs reveal a rapid decrease of the basin depth over the 15-month period from February 2012 to July 2013, during which the depth is reduced by 35 m to a depth of about 25 m. The results show a slower decrease rate after 2015. At the time of the last measurement by CS2 in late 2019, the depth is approximately 15 m, which agrees within the error bars with the height from an ArcticDEM from September 2017. The subglacial lake recharge can be divided into a fast basin uplift of $\sim$ 13 m/yr in the period 2011-2015, and a slow uplift of $\sim$ <1 m/yr in the period 2015-2019.





Figures 3 and 4 show the data available for Lake 2 and Lake 3, respectively. While several TanDEM-X DEMs covering these sites were successfully produced, we were not successful in obtaining any useful CS2 swath-processed data over these subglacial lakes. The collapse basins over Lake 2 and 3 span a smaller area than those over lakes 1 and 4, and they are also
shallower, which could be the reason for the lack of CS2 data here. The early 2011 TanDEM-X observations of Lake 2 show no signs of a collapse basin being formed, however, the four subsequent DEMs (July 2012 to April 2013) clearly show the imprint of a collapse basin. The collapse basin had a depth of approximately 15 m, which did not change over this time span (July 2012 to April 2013), but in the period from April 2013 to December 2013 it appears to fill up completely. After the 2013 melt season, one DEM (TanDEM-X December 26, 2016) shows a collapse-basin feature with a depth of 10 m, while all others
(indicated with light red area in Fig. 3d) show a relatively flat surface at pre-collapse elevations. These observed flat surfaces could be the result of the collapse-basin filling with water, and this was indeed confirmed by optical images close in time to the DEMs in the red box. At Lake 3 (Fig. 4b) the earliest TanDEM-X observations from January 20, 2011, show a surface depression with a maximum depth of approximately 20 m. As it was the case at Lake 2 most DEMs available at Lake 3 show no surface depression after the 2013 melt season (Fig. 4b and d), and only two near-coincident ArcticDEMs from the summer
of 2015 show a small surface depression of approximately 10 m.

Figure 5 shows our results over Lake 4 on the Flade Isblink ice cap in Northeast Greenland. For this collapse basin, the largest of those analysed, several CS2 crossings provide elevation measurements of the collapse basin, and also several TanDEM-X DEMs are available. A TanDEM-X DEM from January and CS2 data from February, 2011 show a flat ice surface in the region of interest. The first data set to observe the collapse basin is CS2 point data from late November, 2011, and it reveals a relatively
flat bottom of a collapse basin with a depth of approximately 95 m. The following data set is also from CS2 data (February, 2012), and it reveals an upward movement of the collapse basin floor of more than 5 m since the previous measurements, three months earlier. As the collapse basin is filled over time, we observe the development of a dome shape at the base of the collapse basin. By April 2017, the last available DEM data (a TanDEM-X), shows that the height of the top of the dome was 20 meters from reaching the pre-collapse surface elevation. The filling rate of the collapse basin changes over time, with a faster
rate of elevation change (∼38 m/yr) immediately after the collapse (November, 2011 - March, 2013) than in the following years, where the rate of elevation change of the deepest point has decreased to ∼11 m/yr (for the period after March 2013). To maintain a visually clear plot, not all data sets are shown in Figure 5b. Figure 5d shows the temporal evolution of the deepest point of the collapse basin until 2021. CS2 swath-processed data after 2017 indicate that Lake 4 drains again in the summer period of 2019, creating a negative elevation change of ∼12 m. The lake seems to quickly recharge during the melt season of
2020.

## 5.2 Subglacial Lake Volumes

Following the methodology outlined in Sect. 4.3 we estimate the temporal changes in the subglacial lake storage. Figure 6a shows the estimated volume changes of Lake 1. We find that the lake had a volume of 0.013 km$^3$ ± 0.001 km$^3$ prior to the drainage event assuming that the lake completely emptied during the drainage (i.e. reaching a volume of 0 km$^3$). After the



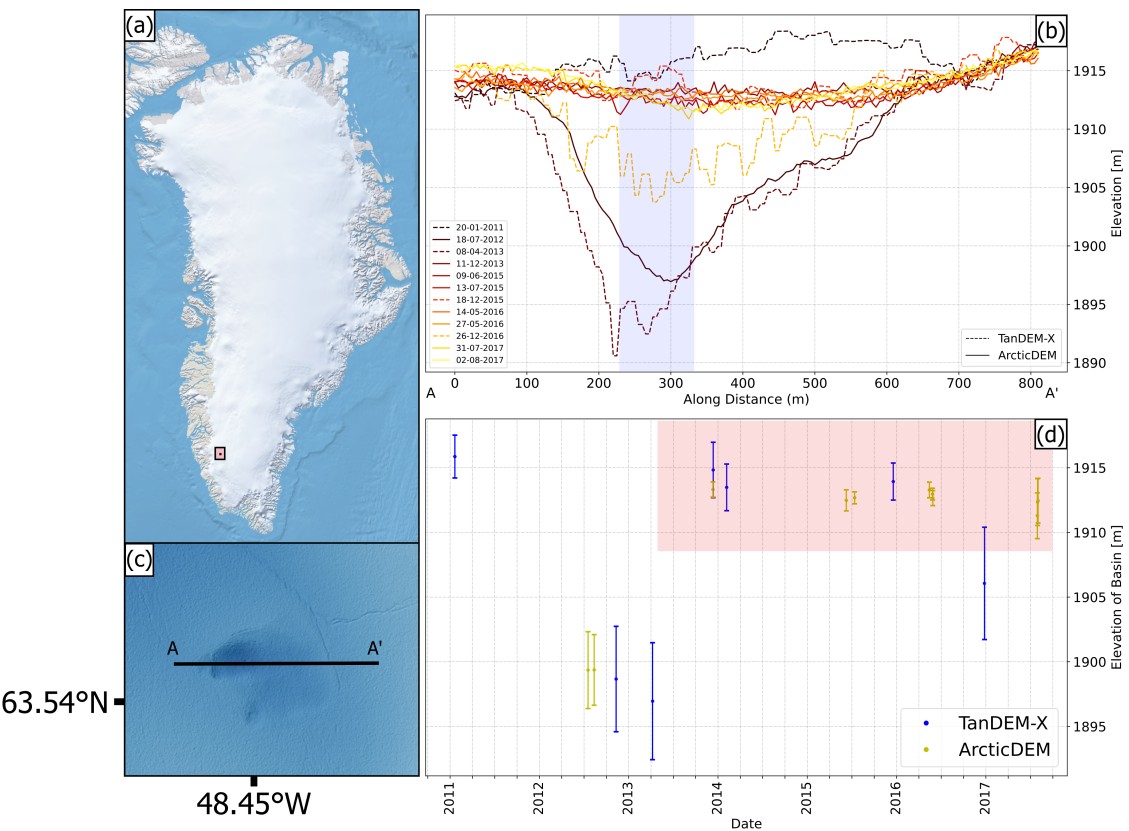

**Figure 3. Lake 2 :** (**a**) The location of the collapse basin. (**b**) Elevation profiles from the two aligned data sets, TanDEM-X is the dashed line and ArcticDEM is the solid line. (**c**) The collapse basin as seen in an ArcticDEM, 18th of July 2012. Line A-A' is the profile shown in (b). (**d**) Time series of the deepest point of the collapse basin from TanDEM-X (blue) and ArcticDEM (yellow).

drainage event in summer 2011, a rapid recharge occurs in the 2012 melt season reaching a volume of 0.008 km$^3$ $\pm$ 0.004 km$^3$. The lake volume stagnates at $\sim$0.010 km$^3$ $\pm$ 0.002 km$^3$ after 2014.

     Figure 6b shows the volume change calculations for Lake 2. The lake had a maximum volume of $\sim$0.0006 km$^3$ $\pm$ 0.00005 km$^3$ in January, 2011. After the drainage in 2011 or 2012 we see a slow recharge, culminating with a lake volume of $\sim$0.0003 km$^3$ $\pm$ 0.0003 km$^3$ in December, 2016. Figure 6c shows the volume calculations from Lake 3. We do not have data before 300   the drainage, and we therefore assume that the lake volume starts at 0 in January 2011. Our data show a slow recharge over the next four years reaching a volume of $\sim$0.0028 km$^3$ $\pm$ 0.00005 km$^3$. Lake 4 has a maximum volume of $\sim$0.3 km$^3$ $\pm$ 0.02 km$^3$ before the lake drained in the summer period of 2011 (Fig. 6d). The lake recharged to half its volume in 2 years reaching $\sim$0.15 km$^3$ $\pm$ 0.02 km$^3$ in 2013. The recharge then slowed down reaching $\sim$0.25 km$^3$ $\pm$ 0.02 km$^3$ in 2018. The lake partially drained again in the summer period of 2019, shrinking to a volume of $\sim$0.21 km$^3$ $\pm$ 0.015 km$^3$, before recharging to $\sim$0.25 305   km$^3$ $\pm$ 0.015 km$^3$ again in 2021.



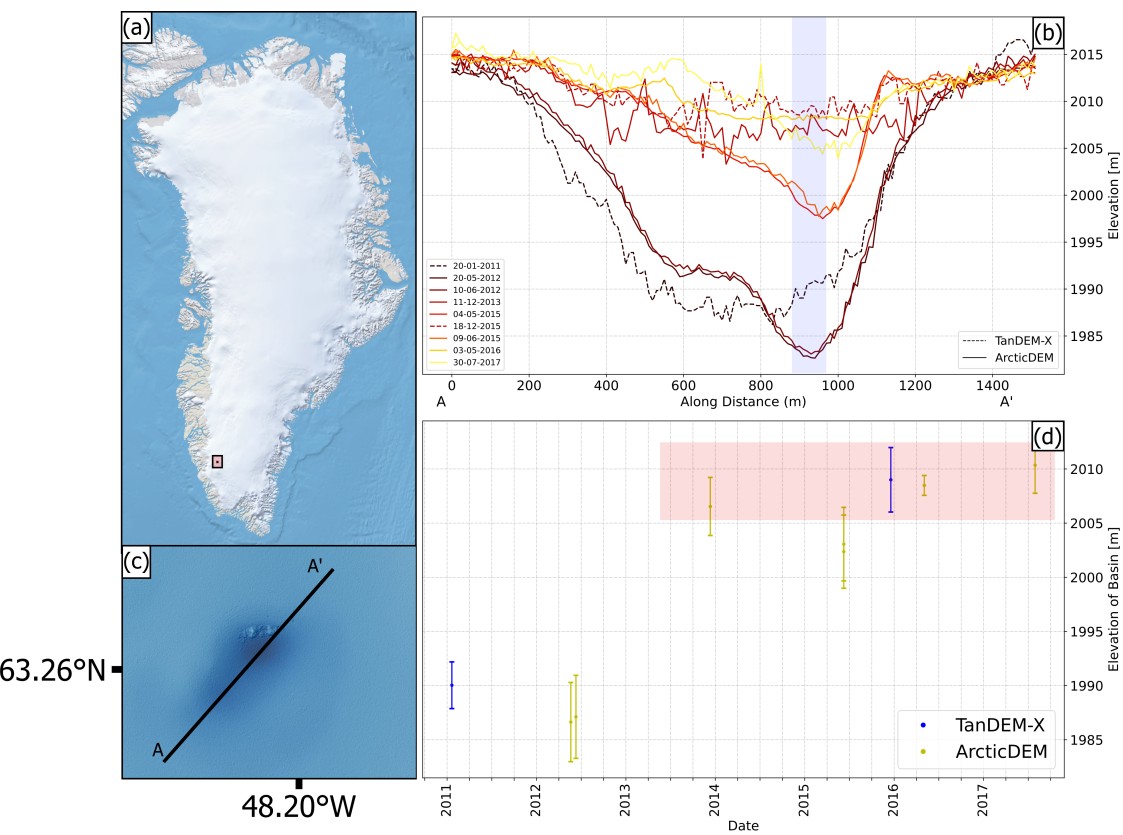

**Figure 4. Lake 3 :** (**a**) The location of the collapse basin. (**b**) Elevation profiles from the two aligned data sets, TanDEM-X is the dashed line and ArcticDEM is the solid line. (**c**) The collapse basin as seen in an ArcticDEM, 20th of May. Line A-A' is the profile shown in (b). (**d**) Time series of the deepest point of the collapse basin from TanDEM-X (blue) and ArcticDEM (yellow).

# 6 Discussion

## 6.1 Lake 1

Over Lake 1 the swath-processed CS2 data and the TanDEM-X data provide new insight into the temporal evolution of the collapse basin. The CS2 data agree within the error margins with near-coincident ArcticDEM and TanDEM-X elevations, giving us confidence of its validity. The availability of the CS2 and TanDEM-X data greatly increases the temporal resolution of data. Compared to previous studies (Palmer et al., 2015; Howat et al., 2015) based on optical imagery with less than one observation per year we are now able to extract multiple observations for each year, and our results confirm the timing of the drainage event of the subglacial lake. Notably, the addition of CS2 observations during the winter 2011/2012 allows us to conclude that no significant recharge of the subglacial lake occurred during this period. The fact that the rate of recharge increased during the following summer confirms the hypothesis proposed by Palmer et al. (2015) and Howat et al. (2015) that





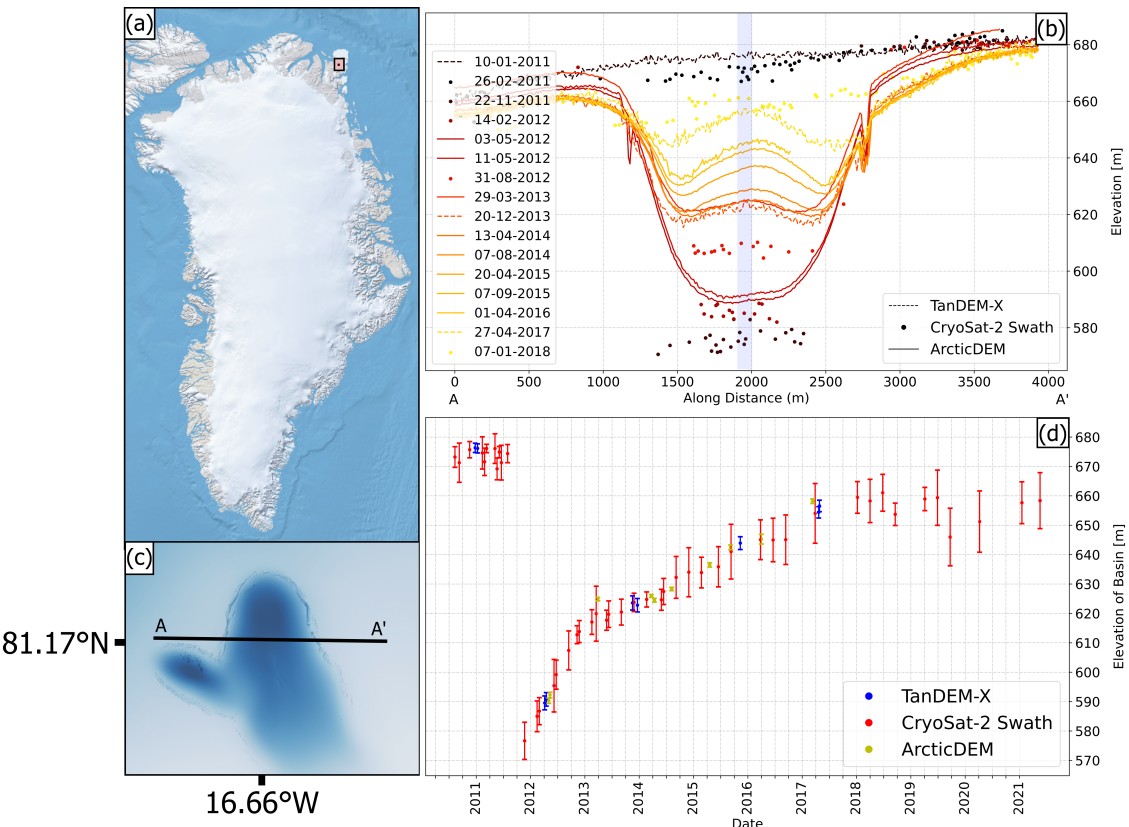

**Figure 5. Lake 4 :** (**a**) The location of the collapse basin. (**b**) Elevation profiles from the three aligned data sets, TanDEM-X is the dashed line, CryoSat-2 is the dotted line data and ArcticDEM is the solid line. **c**) The collapse basin as seen in an ArcticDEM, 3rd of May 2012. Line A-A' is the profile used in Figure (b). (**d**) Time series of the deepest point of the collapse basin from TanDEM-X (blue), CS2 (red) and ArcticDEM (yellow).

the subglacial lake is primarily driven by surface meltwater drained to the bedrock through moulins during the melt season. We further hypothesize that the infilling of the collapse basin after 2014/2015 is likely primarily caused by snowfall and ice flow, and not by recharging of the subglacial lake. This is supported by the observed decrease in filling rate of the collapse basin after 2013, and the fact that the center of the collapse basin moves away from the subglacial lake as a result of local ice flow. This also agrees with model estimates of basal melt rates and subglacial catchment size indicating that small volumes of subglacial water flows into the site of the subglacial lake (see Append A). This kind of glacier response was further modelled and investigated by Aðalgeirsdóttir et al. (2000), where the vatnajökull ice cap in Iceland showed a similar modelled response after a subglacial drainage event created a surface depression.





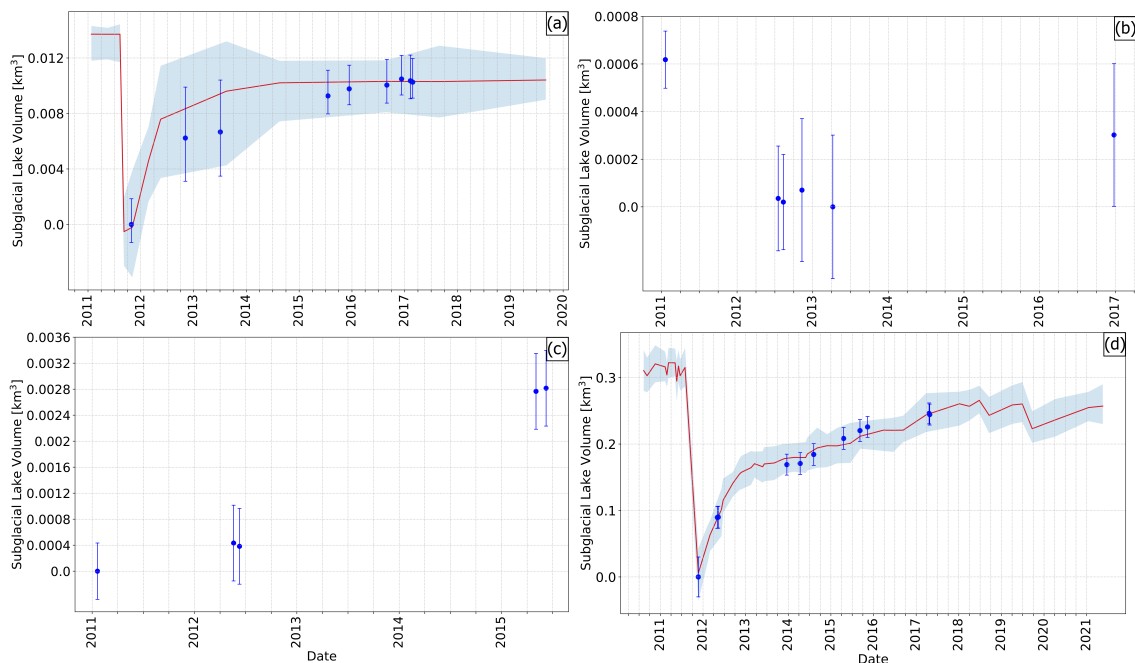

**Figure 6. Subglacial lake volume :** (**a**) time series of Lake 1 volume estimates shown with the red line, and DEM estimates with blue points. (**b**) Lake 2 volume estimations based on DEMs, shown in the blue points **c**) Lake 2 volume estimations based on DEMs, shown in the blue points. (**d**) time series of Lake 4 volume estimates shown with the red line, and DEM estimates with blue points. We assume that the lakes drain completely, and the lake volumes are therefore set to zero immediately after the first drainage event at each site.

## 6.2 Lakes 2 and 3

It was not possible to successfully obtain any CS2 swath-processed data over Lake 2 and Lake 3. This is likely due to their small size (collapse basin area less than 0.6 km$^2$) that does not allow for adequate waveform signals with a coherent phase difference. However, with the availability of several TanDEM-X DEMs covering both lakes, we are able to augment existing observations from Bowling et al. (2019) and gain further insights into their temporal characteristics. Previous studies that relied exclusively on ArcticDEMs could not conclude on the timing of the drainage event over Lake 2 (Bowling et al., 2019), but

the addition of TanDEM-X scenes reveals that the drainage event did not occur earlier than January 2011. Optical imagery indicates that surface water is present in the collapse basins several times after the subglacial lakes drain. At lake 2 Landsat-8 imagery from sprint/summer 2015 shows first a flat surface followed by water presence, indicating a frozen supraglacial lake that melts, which then drains in the summer period of 2016. This is clear in Figure 3 where a flat surface is present in the elevation profiles in 2015, which then shows a collapse basin in the TanDEM-X profile from late 2016.

At lake 3 Landsat-8 images from autumn 2013 and spring 2014 show a flat surface, also suggesting a frozen supraglacial lake, which then melts and drains during summer 2014. This is also clear in the elevation profiles in Figure 3 where the acquisition from late 2013 shows a flat surface, but then in spring 2015 the collapse basin is visible again. Based on this





finding, we suggest that the elevation profiles from the DEMs highlighted with a light red box in both Figure 3(d) and 4(d) map the height of a supraglacial lake forming in the surface depressions, rather than the actual surface depressions. Disregarding those estimates that we believe are associated with surface water, we conclude that the collapse basins over Lake 2 and Lake 3 have not completely filled up by the latest available measurements (late 2016 for Lake 2 and mid 2015 for Lake 3).

### 6.3 Lake 4

Willis et al. (2015) reported that collapse basin over Lake 4 had a depth of approximately 70 m on May 3rd, 2012, and TanDEM-X DEMs processed for this study indeed confirm this estimate (Fig. 5b). However, with the inclusion of CS2 data from November 2011, we can furthermore conclude that the collapse basin has been at least 95 m deep prior to May 2012. In fact, we suggest that the collapse basin likely was more than 100 m deep at the time of the collapse. We arrive at this depth by using the observed average rate of infilling during the period November 2011 to March 2013, and then assuming that this infilling rate is representative for the period from the collapse in August/September 2011 (Willis et al., 2015) to our first post-collapse measurement in November 2011.

Willis et al. (2015) also showed an uplift rate of ∼9 m/yr based on three near coinciding ArcticDEM scenes in May 2012. With the inclusion of CS2 in the analysis, we find that the uplift rate appears to be significantly larger, as we find a relatively constant uplift rate of ∼32 m/yr in the period November 2011 - end 2012. This indicates that a significant level of lake infilling happened within a year of the drainage event indicating that meltwater is readily available at this site. The lake appears to fill also during winter months, indicating that at least some of the input is basal meltwater.

The CS2 data also observed a ∼15 meters drop in elevation (Fig. 5(d)) between May 31 and August 24, 2019, indicating that Lake 4 partially drained a second time, but that the event is substantially smaller than that in 2011. The surface lowering in 2019 is also documented from ICESat-2 data by Liang et al. (2022), who identified it as a drainage event which shortly affected the local horizontal ice velocity. Further investigations into available melt water sources and local ice cap settings are necessary in order to understand why Lake 4 did not drain completely in 2019.

### 6.4 Data Limitations

We are able to retrieve CS2 swath-processed elevation data over Lake 1 and Lake 4 prior to drainage, immediately after the drainage, and during the following recharge. In Figure 2(d) and 5(d) the CS2 swath data provides better temporal coverage than TanDEM-X and ArcticDEM just after the collapse. When applying the CS2 swath processing, careful filtering based on coherence, power and bin number is done. We speculate that when the surface depressions are filled over time, the signal from the bottom does not stand out as clearly in the waveform because it is located closer to the surface returns in the waveform. This could indicate that CS2 swath processing is useful to detect primarily the bottom of only the relatively deep surface depressions with a sizable area, and may explain why it was not possible to obtain data over Lakes 2 and 3, which are not as deep or large as Lake 1 and 4. The fact that the bottom of the collapse basins is flat at Lake 1 and Lake 4 further aids the elevation retrievals immediately after the collapse, because the flat bottom makes the surface a better reflector. Due to the need for careful filtering and tuning of the CS2 swath processor, the CS2 data is not ideal for finding and locating subglacial lake





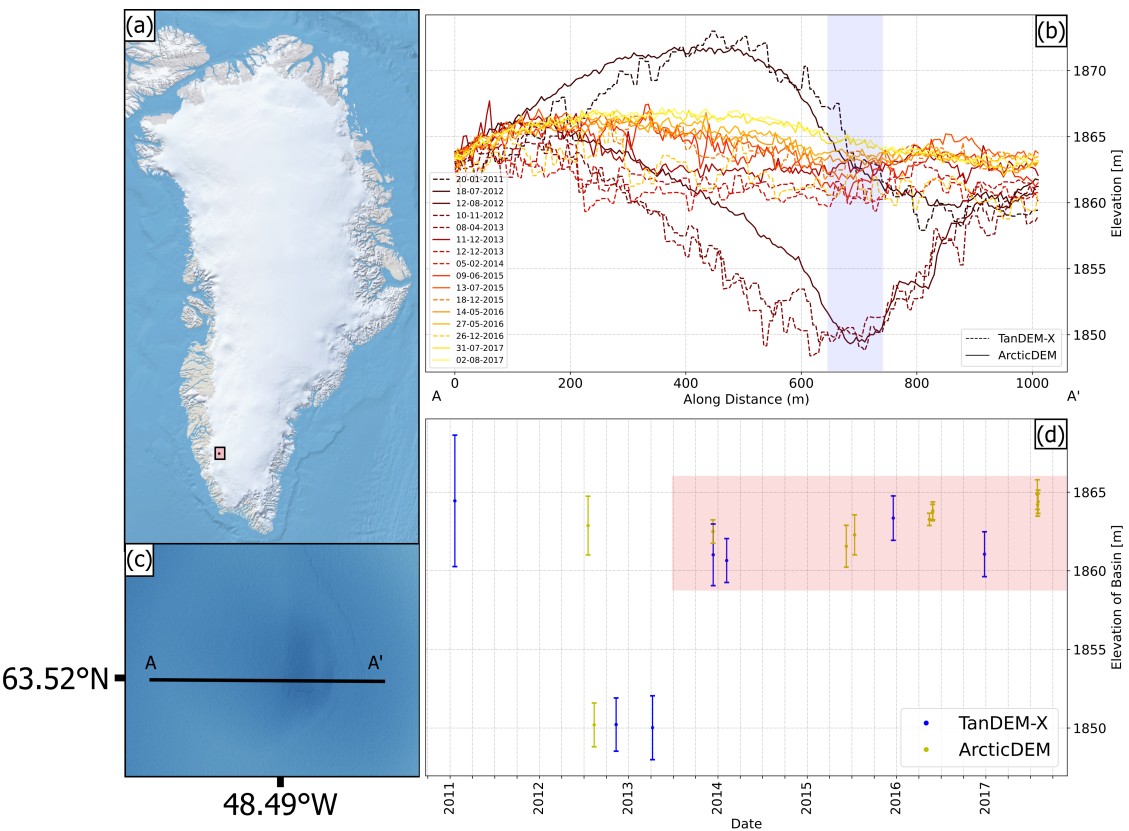

**Figure 7. Possible active subglacial lake:** (**a**) The location of the collapse basin. (**b**) Elevation profiles from the two aligned data sets, TanDEM-X is the dashed line and ArcticDEM is the solid line. (**c**) Possible collapse basin as seen in an ArcticDEM, 12th of August 2012. Line A-A' is the profile used in Figure (b). (**d**) Time series of the deepest point of the lake basin from TanDEM-X (blue) and ArcticDEM (yellow).

activity, since the analysis is dependent on available DEMs. We find that another limiting factor for the success of CS2 swath processing for subglacial lake mapping is that the satellite track must pass directly over the surface depression for bottom echoes to be retrieved. If the surface feature is located too far off-nadir, we do not obtain any valuable data. This limits the use of CS2 to very specific cases of subglacial lake activity, however as seen above, we do find useful satellite crossings for
well-developed collapse basins.

The interferometric X-band elevation data from TanDEM-X suffer from penetration of the X-band SAR signal into the snow and ice surface by several meters, which depends on the properties of the ice (e.g. density, grain size, and dielectric properties) and on InSAR geometry parameters (Abdullahi et al., 2019). Here, we circumvent the penetration bias effect by examining solely height differences within TanDEM-X DEM scenes. Also the CS2 signal may penetrate into the snow, with a penetration
depth different from that of the X-band data. To minimize the elevation changes caused by different surface penetration as well as the elevation changes caused by surface mass balance, we vertically align all the data sets at the rim of the collapse basins



(Sect. 4.1). This implies an assumption that the surface mass balance and the ice properties within the collapse basin are the same as at the rim. This is likely associated with an error since e.g. snow conditions in the depression may differ from those on the rim. At present we do not include this error in our estimates, since we do not have the means to quantify it.

The high-resolution data sets gathered here also give insights into the physical processes driving the refilling of a subglacial lake collapse basin. At Flade Isblink ice cap (Lake 4), we observe a dome forming in the central part of the collapse basin as it refills. This formation suggests a highly active hydrological system at the base of the ice cap, where the low pressure of the central parts of the collapsed basin causes an influx of meltwater which exerts sufficient force to push the central part upwards.

We do not include ICESat-2 data satellite laser altimetry in this study as the main goal has been to densify the time se-
ries covered by the CS2 mission, but we acknowledge that this sensor provides an obvious dataset for future monitoring of subglacial lake activity.

## 6.5  Basal Melt Flux

The derived subglacial lake volume changes can be compared to calculations of basal melt to conclude whether the lake refilling is actually driven by surface water or basal melt. As an example, we calculated the theoretical basal melt across the
predicted upstream catchment of Lake 1, following the method described by Karlsson et al. (2021). The result can bee seen in Fig. A1. The drainage basin for the lake is small and we find that the basal melt flux into the lake is too small (of the order of $10^5 \mathrm{m}^3$/year) to explain the rapid recharge of the lake that we observed. The calculation of the basal melt is uncertain, and the bedrock topography might not be known in sufficient detail to conclude on the shape of the catchment. Even so, the estimates suggest that the subglacial lake is more likely to originate from surface melt water.

## 400  6.6  Hydrologically Connected Subglacial Lakes

In our analysis of Lake 2 and 3, we identified an interesting signal on the ice sheet, which to our knowledge has not been described elsewhere. Located only 3 km southwest of Lake 2, we found indications of a new active subglacial lake. The location, as well as the elevation profiles from ArcticDEMs and TanDEM-X DEMs are shown in Figure 7. The data show a dome-shaped feature on the surface in January 2011 and July 2012. Between July 18, 2012, and August 12, 2012, a rapid
surface elevation change occurs and a feature resembling a collapse basin is formed. Landsat 7 imagery (Fig. B1) confirms the existence of this feature and details its formation between two scenes taken on July 25, 2012, and August 26, 2012, revealing a likely drainage event occurring over the span of maximum 18 days. This drainage event coincides with the unprecedented melt event in July 2012 across the GrIS (Nghiem et al., 2012). The low point of the surface depression is shifted horizontally compared to the dome. As this newly discovered subglacial lake is located approximately 3 km downstream of Lake 2, we
suggest that this lake could be hydrologically connected to Lake 2. The timing of the event further supports this, with the TanDEM-X DEM showing a flat surface at Lake 2 in January 2011 (Fig. 3b), and Landsat-7 imagery further confirms that no collapse basin was present in the summer period of 2011 or in early June of 2012 (Fig. B2). The following ArcticDEM and Landsat-7 image from July 18, 2012, both show a distinct collapse basin after the draining event (Fig. 3 and B2). Therefore, we hypothesize that Lake 2 drained between June 9 and July 18, 2012, and then prompted the draining of the subglacial lake



3 km downstream between July 25 - August 12, 2012. This is to our knowledge the first evidence of hydrologically connected subglacial lakes in Greenland, indicating that water is transferred from one lake to another following a draining event. The small size of the investigated subglacial lakes makes it difficult to assess routing pathways at the bedrock considering the limited spatial resolution of available data sets like BedMachineV4. Hydrologically connected lakes have already been documented in Antarctica (Wingham et al., 2006; Smith et al., 2017), and their influence on the subglacial system has been investigated

(Malczyk et al., 2020).

## 7   Conclusions

The importance of subglacial lakes to the hydrology of the Greenland ice sheet and surrounding glaciers and ice caps is not well understood, largely due to a lack of observations. In this study, we have investigated the elevation changes over four Greenlandic subglacial lakes, which previous studies have identified by the presence of a collapse basin at the ice surface. We

demonstrate how the inclusion of CS2 swath-processed data and TanDEM-X DEMs in addition to ArcticDEMs improve the mapping of subglacial lake activity in Greenland, demonstrating that these and similar data sets should be included in future analyses in order to increase the temporal resolution of the observational records. The small size of the subglacial lakes in Greenland provides a challenge for satellite radar altimetry, and we are only able to recover useful CS2 data over the two largest collapse basins. The TanDEM-X mission provides a valuable additional elevation data source for all lakes throughout

the entire investigated time span (2011-2018). Both TanDEM-X and CS2 data agree well with the ArcticDEM data when we vertically align them at the rim of the collapse basins, and it gives us confidence that they are indeed reliable.

     The use of TanDEM-X DEMs and CS2 data significantly increase the sampling frequency over the four subglacial lake sites. Over Lake 1, the addition of CS2 data during the winter 2011/2012 showed that no significant recharge of the subglacial lake occurred during this period. Previous literature did not conclude on the timing of the drainage event over Lake 2, but TanDEM-

X scenes revealed that it had not drained in January 2011. CS2 data show that the initial depth of the collapse basin over Lake 4 is $\sim$ 35 % deeper (approximately 95 m in late November 2011) than previously found based on ArcticDEMs, and that the filling rate of the collapse basin changes significantly over time. Finally, we identified a signal which we argue is evidence of a previously undetected active subglacial lake in the vicinity of Lake 2. The improved temporal resolution also indicates that this new lake may be hydrologically connected to its upstream neighbour Lake 2, making this the first discovery of hydrologically

connected lakes in Greenland.

## Appendix A:  Modelled basal flux and melt rates

The local basal conditions for Lake 1 is presented in Fig. A1. Here, we use the estimates from Karlsson et al. (2021) and calculate the lake catchment based on local surface and bed topography, following the method from that study. We find that the catchment (black dashed lines in Fig. A1) is fairly small extending only 20 km upstream of the lake location. The flux of basal

melt water into the catchment is approximately $5*10^4$ m$^3$/year and local basal melt rates are of the order of 1-2 cm per year.



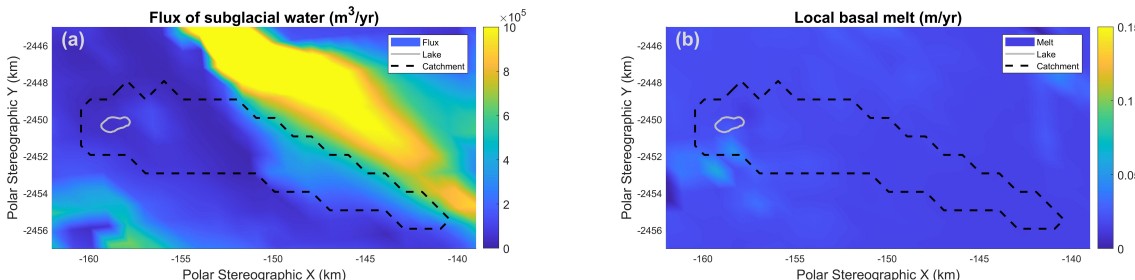

**Figure A1.** Modelled basal conditions for Lake 1 (grey lines) and its catchment (dashed black lines). (a) The flux of subglacial meltwater. (b) Basal melt rates.

## Appendix B: Optical observation of the fifth lake

In the area of Lake 2, we used Landsat-7 imagery to establish if the timing of the draining is connected to the lake which is observed approximately 3 km downstream.

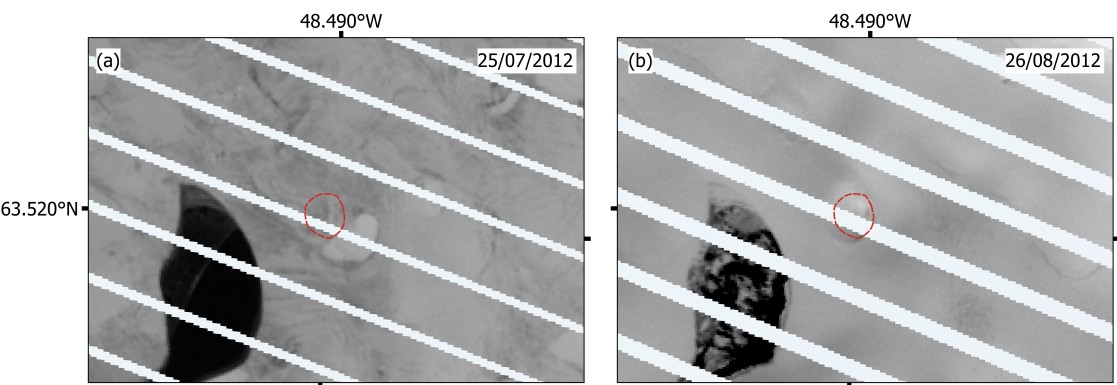

**Figure B1.** Landsat-7 images taken on July 25, 2012, June 9 and August 26, 2012 of the surface before and after the **Lake 5** collapse basin had formed. The ArcticDEM observed August 12, 2012, created the red outline that bounds the collapse basin.

*Author contributions.* LSS and RB planned the study. LSS, RB and SBS developed methodology. NG, AH, NHA and RB carried out swath
processing of CS2 data. NBK calculated basal melt flux. AMS analyzed ice velocities. BW calibrated and delivered the TanDEM-X data. MM, AL, JB, MM and JM contributed to the analysis and discussion of ArcticDEMs. LSS and RB wrote the manuscript with input from all the co-authors, who all discussed and revised the manuscript, and all authors have read and agreed to the published version of the manuscript.





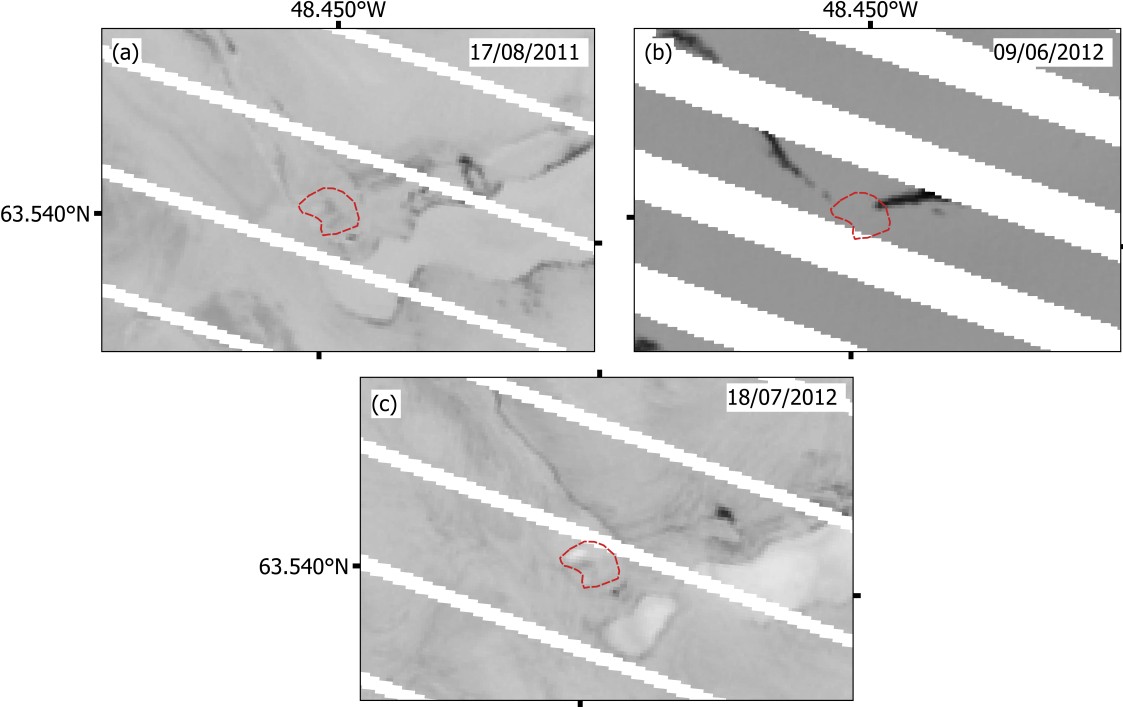

**Figure B2.** Landsat-7 images taken on August 18, 2011, June 9 and July 18, 2012 of the surface before and after the **Lake 2** collapse basin had formed. The ArcticDEM observed July 18, 2012, created the red outline that bounds the collapse basin. On June 9, 2012, it is observed that a body of water partially intersects with the red outline , indicating that a local depression had not yet formed in June 2012. On July 18, 2012, a depression or a peak is observed inside the red outline, which is confirmed as the collapse basin in Figure 3.

*Competing interests.* Some authors are members of the editorial board of The Cryosphere. The peer-review process was guided by an independent editor, and the authors have also no other competing interests to declare.

*Acknowledgements.* This study was carried out in the project POLAR+ 4DGreenland project (2020-2022), which was funded by the European Space Agency (ESA) via ESA Contract No. 4000132139/20/I-EF. MM was additionally supported by the UK NERC Centre for Polar Observation and Modelling, and the Lancaster University-UKCEH Centre of Excellence in Environmental Data Science. The TanDEM-X data was provided by the German Aerospace Center (DLR) via TanDEM-X CoSSC proposal XTI_GLAC7335. We used Ice velocity data from: Joughin, I., B. Smith, I. Howat, and T. Scambos. 2015, updated 2018. MEaSUREs Greenland Ice Sheet Ve-
locity Map from InSAR Data, Version 2, 0478 Boulder, Colorado USA. NASA National Snow and Ice Data Center Distributed Active Archive Center. doi: https://doi.org/10.5067/OC7B04ZM9G6Q. [20190807]. Ice velocity maps were produced as part of the Programme for Monitoring of the Greenland Ice Sheet (PROMICE) using Copernicus Sentinel-1 SAR images distributed by ESA, and were provided by the Geological Survey of Denmark and Greenland (GEUS) at http://www.promice.dk The Arctic DEM strips were downloaded from https://www.pgc.umn.edu/data/arcticdem/



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
