# Peer review of "Improved Monitoring of Subglacial Lake Activity in Greenland."

_The Cryosphere, 2022_

## Author Comment (AC1)

Referee #1

*We thank the referee for their feedback on our manuscript, and suggestions for improvements.*
*In the following, we will reply in detail to all issues raised by the referee and explain how we will revise our manuscript accordingly if we are invited to submit a revised version of it.*
*We show the referee's comments in black and our response in blue italic text.*

This manuscript uses a combination of Cryosat-2 laser altimetry and DEMs from SAR and optical measurements to provide detailed measurements over four previously identified subglacial lakes in Greenland, and one prospective, but not previously identified lake. It provides some details of a set of techniques for combining measurements from these sensors, and offers a longer time series of elevation changes for the lakes than previous studies did, with somewhat more temporal detail. The use of Cryosat-2 data allows the authors to measure the depth of the lake under Flade Isblink immediately after its drainage, and finds a depth for the collapse feature that is significantly deeper than that measured in previous studies. I had trouble identifying the scientific questions that the study answered. Since four of the lakes had been identified in previous studies, the fact of their existence is not news, and the behavior documented in this study is not especially surprising.

*The objective of our analysis was never to document any surprising behavior of the subglacial lakes investigated. Contrarily, we wanted to investigate whether CS2 SARIn data and TanDEM-X data can be used to improve monitoring of subglacial lake activity in Greenland, and therefore, we chose those lakes that were already described in the literature as these provided the possibility to benchmark our data.*
*By documenting that these data are actually useful for monitoring the lake activity, they can/should be included in future subglacial lake studies.*
*That being said, in those cases where the CS2 data actually provides new information about timing or variability, we have explained this.*

The fifth, potential lake identified here is extremely small and is close to one of the previously known lakes, so I am not sure what significance I should attach to its existence.

*We agree with the referee that this lake is small, and we do not claim that it will have great importance in the overall hydrological system or in the runoff from that basin. In spite of its small size, we do think that it is important to document our findings since the active subglacial lake activity is one of the very few ways of actually observing what is happening beneath the ice sheet. We also think that the fact that two lakes might be connected is interesting since this can provide some information about the hydrological pathways.*

The study may be interesting to researchers with a deep knowledge of, and interest in, the particular subglacial lakes studied here, but I am not sure how wide this audience is likely to be.

*We are sorry to learn that the referee thinks that this study will not be interesting to a larger audience. We do, however, not share that point of view. For the entire scientific community that works on subglacial lakes/hydrology, we believe that it is an important conclusion that additional datasets can be used to improve future monitoring efforts.*

The authors suggest that measurements over subglacial lakes have the potential to inform our understanding of subglacial water flow, but I really didn't see much development of this potential in this study. The abstract identifies the demonstration of techniques as a goal of the study, but the technical discussion of the techniques is brief and the presentation of the measurements is not very detailed.  I would recommend reworking the study, either to focus on how each of the techniques performed at lake 4 (which had very large relief and elevation change) and at lakes 2 and 3 (which were small, and where the Cryosat-2 data didn't work well), or to try to better understand the implications of the measurements for the subglacial hydrology of the ice sheet.

*We see that referee #2 also states that it would be beneficial to rework the manuscript to make the objective clearer. We suggest revise the manuscript to include more information on the data, including uncertainties, quality and methods. We will include more figures of the data, inclusing waveforms from different tiems over a chosen lake, and spatial plots of data coverage for each processing step.*

*We suggest removing the basal melt calculation and associated discussion.*

Line 34:  Should note that this possibility was investigated in some detail by other studies (Stearns 2008, https://www.nature.com/articles/ngeo356)  (Smith et al, 2017 (cited in the  manuscript)  And (Zwally and others, 2002, https://www.science.org/doi/10.1126/science.1072708), and that  net dynamic changes after very large water inputs were negligible.

*We assume here that the referee is referring to Lines 32-34 and the statement that: "The sudden drainage and outburst flood of a subglacial lake might temporarily affect ice flow velocities downstream from the lake location Palmer et al., 2015; Liang et al., 2022)."*

*We agree with the referee here, which is also why we have written that it might impact ice velocities. We do not agree however that all the papers listed by the referee conclude that the effect is negligible. Contrarily, some quotes from those papers are:*

*"Our findings provide direct evidence that an active lake drainage system can cause large and rapid changes in glacier dynamics." (Stearns et al., 2008 )*
*"The indicated coupling between surface melting and ice-sheet flow provides a mechanism for rapid, large-scale, dynamic responses of ice sheets to climate warming." (Zwally et al, 2002).*

*The Zwally et al (2002) paper focuses on surface melt and not subglacial lakes though, so we do not see the relevance here – even though the surface and basal hydrology are connected.*
*We agree that the Smith et al., 2017 paper describes a case where no connection between drainage and ice velocity is observed, but we do not see how this contradicts our statement in the manuscript.*
*We suggest revising the paragraph in the manuscript to emphasize that some studies show a connection between lake drainage and ice velocity and that one found no connection. We will include the suggested references.*

Line 88: "Classified" is not the right verb here. "Asserted" might be better
*We agree and will revise accordingly*

Section 3-1: Is there any way the selection of thresholds can be formalized? The thresholds selected here seem ad hoc, and it would be useful to discuss how they were chosen.
*We agree that the threshold selection seems ad hoc. In our study, we did try to find a more formalized approach but did not succeed. We find that the threshold is very case-specific and is dependent on e.g., surface conditions (scattering properties), the geometry of the satellite orbit versus lake location, and the geometry of the surface depression.*
*We suggest revising sect 3.1 in the manuscript to explain this more clearly.*

Line 140: "highly dynamic" should be "rough"
*We will rephrase it to "highly variable", as it is not only defined by its roughness.*

Line 141: Is the incoherent component in the processing, or in the radar reflections?
*We will rephrase it to "… in a larger incoherence in the data."*

Line 145: should be "assumed to be representative"
*We will rephrase it to "is assumed to be representative"*

Line 145: "were deemed as errors" should be "were assumed to be errors"
*Agree. We will revise accordingly.*

Line 146: "Across swath tracks close to the basin rim" should be "swath-processed data from tracks close to the basin rim"
*Agree. We will revise accordingly.*

Line 148: remove commas around "which is removed"
*Agree. We will revise accordingly.*

Line 183 "vertical alignment" should be "vertical offset"
*Agree. We will revise accordingly.*

Line 184: delete "found to be"
*Agree. We will revise accordingly.*

Line 197: "but we see" should be "and we see"
*Agree. We will revise accordingly.*

Line 201: "such as" -> "including"
*Agree. We will revise accordingly.*

Line 211: What is the basin shapefile?
*Here we refer to the manually delineated basin outline. We will revise to make this clear.*

Figure 1 (and all similar figures)
*We assume that this comment is actually about figure 2,3,4,5, and 7*

The map extent is too broad to give a useful context for the lake location.  Should instead show a context map with the regional topography and the locations of adjacent glaciers in some detail, with a reference map in a separate figure to indicate the locations of figures 1-7
*We agree that these figures can be improved. We suggest making one common figure to show all the locations:*

[Figure]

Need to provide a color bar for panel c

*Agree, we will add this to the new figure (see above).*

The yellow lines in panel b are very hard to see
*We agree that they are hard to see, but we have tried to plot this in many different ways .*
*We will revise the figure to make the yellow lines more clear.*

The range of contrast in the colors in panel b does not really allow the distinction between different CS2 dates.  Different symbols should be used to denote different dates.
*We will revise the figure to include different symbols*

The legend should explain the blue shaded bar
*We will revise accordingly*

*Suggested figure revisions:*

[Figure]

Line 224: Subtracting the median height does not make sense, as the offset subtracted is will depend on the height distribution of the rim.   It would be better to subtract a median height anomaly relative to some reference DEM.  Is this what the authors mean to say?

*We agree that it is indeed an approximation to use the median height along the rim, and that it will have an impact on the absolute volumes.  A more accurate method would be to reference surface (plane) based on the rim elevations.*
*We will investigate if this would change the results and revise the methods/manuscript accordingly.*

Line 226: "cubing the 2sigma": What is this, and why does it give an error estimate?  This needs much more detail to explain and/or justify what is done here.

*We will rewrite the paragraph "line 226" to:*
*"To estimate the error of the DEM volumes, we compute a new set of volumes at each grid point, with the uncertainty from the depth estimation for the used DEM added to the extracted surface elevations. We then subtract the previous set of volumes, and sum the discrepancies, to get the total volume error at each DEM."*

Line 228: Need to specify which depths and volumes are used here, and need to connect these, using consistent terminology, with the depths derived from the DEMs and from CS2.  Are "the depths" referenced here the depths of the deepest point from CS2?

*Yes, the depths here are the deepest points. We see that clarification is needed and we will revise this section.*

Line 231 / equation 1.   How does the derivation of R and V take the error bars into account?  More detail is needed.

*We agree that a further explanation of how the uncertainty impacts the derivation of R and V is needed.*
*We have changed this paragraph:*
*"Since the collapse basin shape and form changes over time, the shape factor R changes over time, and we fit a smoothed function, R̃(t), through all available R(ti) values, taking their error bars into account."*

*To:*

*" Since the collapse basin shape and form changes over time, the shape factor R changes over time, and we fit a smoothed function, R̃(t), through all available R(ti) values, using a least squares method.*

*Because of the uncertainty of the depth and volume estimates, there is not one unique solution, making this an ill-posed problem. To account for the uncertainty, we introduce a regularizarion parameter "α" that penilazes the cost function in the least squares solution, to prevent overfitting to the ill-posed problem."*

Line 236: It would be useful to demonstrate how R~(t) varies in time based on the available DEM data.
*We do not see how such a figure would improve the manuscript, but we could provide such a figure in an Appendix.*

Lines 225-236: The methodology here does not seem to capture the true uncertainty in depth (and volume) estimates based on the CS2 data. When there is a large spatial variation in elevations in the DEM data, they are assessed a large error based on the slope and roughness within the relevant part of the lake, but CS2 data generally give a small number of elevation measurements at these times, and are assessed a smaller error. Would it not make sense to apply roughness information from the DEMs to the CS2 data to assess their errors?
*We agree that it does make sense to add a basic "roughness" uncertainty to the CS2 uncertainties. We will do so and update figures.*

Line 246: add comma after "coverage"
*We will revise accordingly*

Line 264: It would be useful to explore why CS2 did not provide data over lakes 2 and 3. Were there no footprints that intersected the lake boundary? Was the coherence too low?
*We agree that it would be useful to further explain why that is the case.*
*The preliminary reason for the lack of data is that the surface depressions were too small, it was therefore difficult to differentiate multipeaked waveforms as a reflection from both a depression and a surface. Furthermore, the narrow structure of the depressions also increased the incoherent component of the phase difference, thus making it tough to do a phase unwrapping. We suggest to add figure(s) of some selected radar waveforms to clarify.*

Line 265: Please show the power image from TanDEM-X for early 2011. It would be interesting to know if there are any reflectance features associated with the about-to-drain lake.

[Figure]

*TanDEM-X SAR amplitude image of Lake2 (SouthernLakes) from 20-01-2011 (left) and optical image from Google Earth from 09/2012.*

*Interestingly, the subglacial lake and its northwestward-flowing channel appear slightly darker than their surroundings in the TanDEM-X amplitude data. There are other darker structures nearby, so identifying the subglacial drainage structures based on SAR amplitude alone does not seem sufficient. However, it could help to identify and locate them.*

[Figure]

*left the southern lake (Lake2); right the DEM from 20-01-2011 TanDEM-X acquisition*

Line 279: "CS2 point data" :should this be "CS2 swath data"
*We will revise accordingly*

Volume calculations: Except for Flade Isblink, these volumes are exceedingly small. Compared to lake discharges in Antarctica, they are miniscule, and those Antarctic discharges had almost no effect on ice dynamics. What is the justification for saying that the lakes studied here might be important for ice dynamics?
*As mentioned earlier, there are references for how subglacial lake drainage can affect ice velocities.*

320: Should compare volume-change estimates against surface runoff estimates from (e.g.) RACMO.
*We agree with the referee that a study that includes both estimates of basal and surface melt with the subglacial lake activity would be interesting. This would however require modelling/observations of how much of the surface melt water that reaches the bed, which we believe is outside the scope of the current manuscript. Also, since there is a wide spread in the predicted runoff estimates from different RCMs such a study should include several models (Fettweis et al., 2020).*

358: "shortly" should be "briefly"
*We will revise accordingly*

373: "off-nadir" should be "off nadir"
*We will revise accordingly*

376-384: this repeats material found in the methods section.
*We agree and suggest deleting the sentence from line 376-380, but keeping the last part (380-384) that emphasizes that we do not take the associated error into account.*

378: delete "parameters"
*See comment above*

387: is "highly active" all that can be determined here? This doesn't seem like a lot has been learned.
*As the focus of a revised manuscript will be more on data and methods and less on the geophysical interpretations, we suggest to delete this sentence.*

Section 6.6
To conclude that the activity of the new potential lake affected the drainage of lake 2, the authors would need to present evidence that it is unusual for water to reach the bed in volumes comparable to those discharged by the new lake. Looking at the images in appendix B, it appears that there is abundant water on the surface of the glacier, and it seems likely that this water often drains through moulins. Why, then, should we believe that the drainages of lakes 2 and 3 are anything but coincidental? Even if they were not coincidental, what specifically does this tell us about the hydrology of the glacier bed that we could not have inferred already?
*This is true. Here, we simply want to point to the fact that the timing of the events could imply that they are connected. We do not foresee to do any detailed analysis in this work to support this hypothesis. But we agree that the section can be improved by expanding on the information and discussion. We will do so.*

Appendix A: Why would the basal melt rates be important in this area? Water fluxes from surface melt must dwarf these rates by orders of magnitude. Please consider surface melt first.
*We suggest removing the basal melt plot and associated discussion.*

Appendix B:
Figure B1: Indicate the location of this lake relative to lake 2. Also- what is being mapped here? The difference between panels a and b seems to mostly be that in panel B the surface is covered with snow, while in panel A it is mostly bare ice. The interpretation of the change in the collapse basin is not at all clear to me.
Figure B2: There is a lot of variability in surface conditions between these images. The interpretation in the text is not at all convincing.
*We will revise Appendix B to clarify*

Data availability: I didn't see a statement about data availability for the CS2 swath-mode data.

*We will be happy to make the data available. We will do so on data.dtu.dk and provide the link and information needed in the revised manuscript.*

**References**

*Fettweis, Xavier, et al. "GrSMBMIP: intercomparison of the modelled 1980–2012 surface mass balance over the Greenland Ice Sheet." The Cryosphere 14.11 (2020): 3935-3958.*

*Palmer, S., Mcmillan, M., and Morlighem, M.: Subglacial lake drainage detected beneath the Greenland ice sheet, Nature Communications, 6, https://doi.org/10.1038/ncomms9408, 2015.*

*Liang, Q., Xiao, W., Howat, I., Cheng, X., Hui, F., Chen, Z., Jiang, M., and Zheng, L.: Filling and drainage of a subglacial lake beneath the Flade Isblink ice cap, northeast Greenland, The Cryosphere Discussions, pp. 1–17, 2022.*

*Stearns, L., Smith, B., and Hamilton, G.: Increased flow speed on a large East Antarctic outlet glacier caused by subglacial floods, Nature Geosci, pp. 827—831, https://doi.org/10.1038/ngeo356, 2008 Stearns et al., (2008).*

*Smith, B. E., Gourmelen, N., Huth, A., and Joughin, I.: Connected subglacial lake drainage beneath Thwaites Glacier,West Antarcticaf, The Cryosphere, 11, 451–467, 2017.*

*Zwally, H. J., Abdalati, W., Herring, T., Larson, K., Saba, J., & Steffen, K. (2002). Surface melt-induced acceleration of Greenland ice-sheet flow. Science, 297(5579), 218-222.*

---

## Author Comment (AC2)

Referee #2

*We thank the referee for their feedback on our manuscript, and suggestions for improvements.*
*In the following, we will reply in detail to all issues raised by the referee and explain how we will revise our manuscript accordingly if we are invited to submit a revised version of it.*
*We show the referee's comments in black and our response in blue italic text.*

**General Comments**

This paper combines multiple satellite missions to improve the temporal resolution of ice surface elevation change measurements over 4 previously identified active subglacial lakes in Greenland to provide new constraints on lake volume and evolution. In addition, they find one potential new active lake that might be hydrologically connected to one of the known lakes (although see specific comments). The study is generally well written with some nice figures, and I found the combination of methods to improve the temporal resolution convincing. I did, however, find quite a few minor errors or places which needed further clarification (see specific comments below), and I agree with the other reviewer that the implications of their findings are currently not clear, and could do with expanding / reworking. For example, could you combine your improved monitoring of recharge rates with your basal melt modelling (expanded to all sites), to make this a more significant component to better explore the role of surface vs basal melt. How do your recharge rates/ drainage rates compare to elsewhere? Can you use your improved timings of drainage to better link to triggers?

*As also mentioned in our reply to referee #1 the aim of our study has been to investigate whether CS2 SARIn data and TanDEM-X data can be used to improve monitoring of subglacial lake activity in Greenland. Both referees suggest that the manuscript is reworked to make its aim clearer. We would prefer to keep the focus on this paper on the data and its usefulness in subglacial lake monitoring. We suggest removing the basal melt plot and discussion from a revised manuscript, and instead improve and expand on the data method section and associated discussion.*
*We will further expand on the discussion and conclusion to focus on possible applications of our findings – e.g., that it can be used to better link to triggers.*

**Specific Comments**

L4 – Antarctic Ice Sheet
*Agree. We will revise accordingly.*

L6 – I think it would be worth mentioning earlier in the abstract that active lakes are typically identified from ice-surface elevation changes to put this point into context.
*Agree. We will add the following sentence to the abstract: "Active lakes may be identified by local changes in ice topography caused by drainage or recharge of the lake beneath the ice."*

L14 – It is odd to mention surface hydrology at the end as this is not discussed in the rest of the abstract.
*Agree. We will shorten the last sentence to: "These findings show how improving the measurement capabilities of subglacial lakes, improves our current understanding and knowledge of the subglacial water system."*

L21 – not sure this reference is appropriate here as it focuses on predicting lake locations. Perhaps refer to the Livingstone et al. (2022) study instead.
*Agree. This was a mistake. We will refer to Livingstone et al. (2022) instead of Livingstone et al. (2013).*

L24 – "steeper ice surface slopes"
*Agree. We will revise accordingly.*

L27 – delete "further". Your previous points were around different settings not detection.
*Agree. We will revise accordingly.*

L30 – the use of e.g. in this sentence does not work that well. Can you combine the first part of this sentence with the second part of the next to provide a more general mechanism for lake drainage?
*We suggest revising the sentence to:*
*"The lake will eventually drain when filled with enough water to resist the pressure exerted by the overlying glacial load (Chandler et al., 2013), hence a subglacial lake drainage events can be triggered by a prolonged addition of surface meltwater (Livingstone et al., 2022)."*

L33 – I think Palmer look at vertical displacement, but don't really mention horizontal displacement. It might be better to refer to some of the key velocity studies in Antarctica or Iceland here.
*Agree, we will change reference from Palmer et al., (2015) to Magnusson et al, (2007) and Stearns et al., (2008).*

L35 – suggest – "Notably, the period of lake recharge following a drainage event provides…"
*Agree. We will revise accordingly.*

L49 – should this be InSAR not SARIn?
*We made an error here. We will rephase to: "synthetic aperture radar interferometric (SARIn) mode mode".*

L56 – delete "source of" to avoid repetition of this word.
*Agree. We will revise the sentence to : "An additional source of high-resolution ice surface topographic information is provided by two Digital Elevation Models (DEMs);..."*

L59 -this makes it sound like there have just been 4 active lakes identified in Greenland. There are actually 7 – see Livingstone et al. (2019). Worth noting this and rephrasing to justify the four you have chosen.
*We agree, and will mention the three other subglacial lakes beneath the Isunguata Sermia. We suggest to add this paragraph to Section 2:*
*"We have chosen not to include the three subglacial lakes located beneath the highly dynamic Isunnguata Sermia glacier due to the the small size of the collapse basins and their location very close to the ice sheet margin (Livingstone et al. 2019)."*

L66 – this makes it sound like Bowling et al. (2019) identified all these lakes. It would be better to cite the original papers for all these lakes.
*We agree and we will cite the original papers.*

L72 – Figure 2a is cited before Figure 1.
*We will include a new figure in a revised manuscript. It will show the location of all of the lakes and will be the first figure we will refer to.*

L83 – is this a subsidence event in 2004? Not clear.
*Howat et al, 2015 states that "Drainage occurred in two episodes: a smaller event in 2004 and a larger one in 2011."*
*We will delete the current sentence and revise the current L 76-77 to: "These studies find that Lake 1 has drained both in 2004 (smaller event) and one in 2011 (larger). The 2011 drainage occurred with an unknown rate within two weeks (28 June, 2011 to 12 July, 2011), resulting in the formation of a collapse basin in the ice sheet surface."*

L94 – suggest:  "… which could indicate some recharge of the subglacial lakes by surface water".
*Agree. We will revise accordingly.*

L97 – Flade Isblink Ice Cap
*Agree. We will revise accordingly.*

L105 – clarify whether this was infill of the collapse surface basin or infill of the subglacial lake by surface water causing the ice surface to rise.
*We agree that the paragraph needs to be more specific.  We suggest changing it to:*
*"The elevation of the collapse basin rapidly increased by 30 meters over the following two years due to inflow of surface water to the subglacial lake, and between August 2012 and April 2013 a topographic bulge appeared in the basin (Willis et al., 2015)."*

L107 – can you quantify the elevation change associated with this event?
*Liang et al (2022) observed an elevation change of 10 m. We will add this to the text.*

Section 3 – it would be useful to provide details on the final resolution and vertical/horizontal errors of the processed datasets.
*As mentioned above, we suggest to rework the manuscript to put more focus on the data, and processing methods. We will elaborate on e.g. resolutions and errors.*

L112 – SARIn has already been defined.
*Agree. We will revise accordingly.*

L124 – It would be helpful to quantify this change in density.
*The density of the swath-processed data compared to the POCA varies and depends on e.g. the thresholds used in the swath processor, and the physical properties and topography of the surface, which affects the waveform. We will add the following sentence to the paragraph:*

*"The swath processing leads to a 10 to 100 folds increase in elevation measurement compared to the POCA approach, as the output depends on e.g. processing thresholds chosen and the physical properties and topography of the area."*

L128 – "… thresholds compared to those usually applied …."
*Agree. We will revise accordingly.*

L134-138 – Maybe this is because I am a visual person, but a figure showing the raw to processed data points would be really helpful here in allowing the reader to judge the effectiveness of the approach.
*We agree that such a figure will be good to include. As previously mentioned, we will revise the manuscript to put more focus on the data and processing methods. We propose to include figures to show the data coverage (geographical maps) and quality (2d waveform plots).*

L143 – not sure why you need the word "apparent"?
*Agree. We will delete it.*

L145 – "assumed to be representative"
*Agree. We will revise accordingly.*

L146 – how close in time? This is rather vague and could do with rephrasing.
Rasmus
*Agree. We will rewrite this paragraph to clarify.*

148 – would be useful to get a rough estimate – 1%, 10%, 90%? See my comment above, a figure showing the stages in the processing would be helpful here.
*Agree. We propose to include new figures to illustrate better (see comment above for L134-138). We will also provide the information (%) in the text.*

L160 – "data takes located"? Is there an error here, I didn't follow this part of the sentence?
*We will delete "takes".*

L181 – "the vertical bias."
*Agree. We will revise accordingly.*

L192 – given you correct using local ice flow, it would be useful to know here whether this 100 m in 10 years equates to a local ice flow in this region of ~10 m/yr.
*We are not sure what the referee is asking for here.*

L198 – "One reason for this is that the…" I don't really get the point around the size of the lake as surely it is the lake edge that you are tracking.
*Correct. We will revise the sentence to: "One reason for this can be that the subglacial lake drains again in the observational period, which could make the potential ice flow less evident since the collapse basin is re-formed over the stationary location of the subglacial lake."*

Section 4.2 – are these calculations still based on the local difference compared to the basin rim? If not, could these not be influenced by the different penetration depths etc? I don't really follow the approach to calculating the deepest depths (why take a mean if looking for deepest point) or the use of standard deviation. Is this not just a measure of roughness of the floor of the basin? This needs clarifying.

*These calculations are the absolute elevations of the aligned datasets. Since the datasets all are aligned at the rim (section 4.1) they will not be affected by different penetration depths. We will rephrase to make this more clearly.*
*We agree that the mean and standard deviation is also a measure of the roughness of the floor of the basin, but these will inherently have an impact on the accuracy of the measurements. We will rewrite this part to clarify.*

L226 – In other papers error is calculated by multiplying the internal error of the DEM by the lake area both before and after drainage and adding together in quadrature. What is the basis for your approach, especially as you state that 2 standard deviations is not a true measure of their accuracy?
*We address this in the answer to ref. 1:*
*We will rewrite the paragraph "line 226" to:*
*"To estimate the error of the DEM volumes, we compute a new set of volumes at each grid point, with the uncertainty from the depth estimation for the used DEM added to the extracted surface elevations. We then subtract the previous set of volumes, and sum the discrepancies, to get the total volume error at each DEM."*

L260 – does this actually show uplift? Could you run regression analysis over the two periods to calculate the recharge rates more accurately.
*No, not a significant uplift. We will revise the sentence to:*
*"The subglacial lake recharge can be divided into a fast basin uplift of ~13 m/yr in the period 2011-2015, and a period with no significant uplift from 2015-2019."*

L276 – Flade Isblink Ice Cap
*Agree. We will revise accordingly.*

Section 5.1 – Some of the text in this section would I think be better incorporated into the figure captions e.g., "To maintain a visually clear plot not all data sets are shown in Figure 5b." This would help the flow of this section while making the figures standalone.
*We agree and will revise accordingly.*

L297 – Although this is the maximum volume measured, it is the minimum possible volume (given you might have missed the period of maximum collapse (i.e. it might have collapsed and then recharged between data points).
*Good point. We will revise the sentence to :*
*"The lake had a measured maximum volume of ~0.0006 km3 +/- 0.00005 km3 in January, 2011, although since the timing of the drainage is poorly constrained, the actual maximum volume is likely larger than the observed."*

L310 – can you quantify this – re. number of data points over X years?

*Good point. We will provide more precise numbers for the increase in data points by inclusion of the CS2 and TanDEM-X data.*

L313-315 – There does seem to be some signal of the final collapse and then beginning of the recharge period though during the winter 2011/2012 period. To better test this it would be better to split these components and calculate the recharge for the upwards tick as a rate vs. the summer after.
*What we meant was that we do not see that there is a recharge in late 2011 when taking error bars into account. We will rephrase to make it more clear: "Notably, the addition of CS2 observations during 2011/2012 allows us to conclude that no significant recharge of the subglacial lake occurred in the second half of 2011, while recharge is observed throughout 2012."*

L316 – How do you know it is bedrock? Suggest change to "bed"
*Agree. We will revise accordingly.*

L318 – Ok, but could this not be associated with a decrease in filling over time as the area increases (i.e. for a given melt input the rate will decreases because of the basin shape?). I think it is fairly common for recharge rates to slow over time.
*Good point. We will rephrase the sentences to: "We further hypothesize that the infilling of the collapse basin after 2014/2015 is likely primarily caused by snowfall and ice flow, and less recharging of the subglacial lake due to the fact that the center of the collapse basin moves away from the subglacial lake as a result of local ice Flow. The filling rate will however usually slow over time due to the geometry of the lake."*

L320 – "model estimates of basal melt rates" – ok, based on what data? Need a supporting reference and to quantify. I don't quite see how this point fits with the idea of ice flow/snowfall. What is the surface mass balance change?
*As previously mentioned we suggest to remove the basal melt estimates (Appendix A) and discussion from the manuscript at part of shifting the focus towards the data and methods.*

L321 – Appendix not Append.
*Agree. We will revise accordingly.*

L322 – capitalise vatnajokull ice cap
*Agree. We will revise accordingly.*

L330 – A more positive spin would be to give the time span over which the drainage could have happened.
*Agree. We will revise accordingly.*

L331 – "drains"
*Since we are here discussing two lakes we believe that it should be "drain" and not "drains"?*

L332 – "spring"
*Agree. We will revise accordingly.*

L334 – In which case, how do you know whether Lake 2 is actually a supraglacial lake that is filling and draining? It would be useful to confirm whether the 2011 data is associated with surface water or not.

*Agree. We will investigate and include this in the analysis.*

L347 – please state the infilling rate from this calculation.

*Agree. We will revise accordingly.*

L352 – It would be useful to incorporate the results of Liang et al. (2022) into this discussion as they look at seasonal recharge (i.e. impact of warmer summers on recharge rate, with rates of up to 49 m/yr).

*We will incorporate the Liang et al. (2022) further on our discussion.*

L357-358 – "which shortly affected the local horizontal ice velocity" – needs rephrasing.

*Agree. We will rephrase the sentence to: "The surface lowering in 2019 is also documented from ICESat-2 data by Liang et al. (2022), who also identified it as a drainage event. They also document that this event caused the ice velocity downstream from the lake to abruptly but shortly increase. "*

L368 – "basins are flat"

*Agree. We will revise accordingly.*

L378 – You use SARIn elsewhere.

*Yes, but this is correct. For CryoSat-2 the data is called SARIn.*

L386 – Ok, but others have shown this in their data, so should acknowledge this.

*Agree. We will revise accordingly to acknowledge.*

L387-388 – this needs clarifying? Where is this low pressure region?

*We suggest to delete this sentence.*

Section 6.5 – this seems rather tagged on and is not explained in the methods, and very briefly here. I would suggest introducing earlier in the method section and showing in the results, or deleting.

*We suggest to delete this part to focus more on the data and methods.*

L402 – There is also a new paper in TCD by Fan et al. (2022) with lots of new active subglacial lakes. Is this one of the lakes in their inventory? https://tc.copernicus.org/preprints/tc-2022-122/

*As stated above we will include Liang et al (2022) further in our discussion.*

Section 6.6 – given that this period coincided with a large melt event could these both be responding independently to large volumes of water accessing the bed in this region? More information is needed here to better test this hypothesis.

*This is true. Here, we simply want to point to the fact that the timing of the events could imply that they are connected. We do not foresee to do any detailed analysis in this work to*

*support this hypothesis. But we agree that the section can be improved by expanding on the information and discussion. We will do so.*

L422 – Greenland Ice Sheet.
*Agree. We will revise accordingly.*

L423 – "investigated elevation changes"
*Agree. We will revise accordingly.*

**Figures/ Tables**

Given that you correct based on the elevation outside of the collapse basins, I was surprised to see many of the plots with a y-axis of elevation rather than elevation anomaly (relative to the tie points). It would be useful to clarify whether you use similar time periods as cross-over points or how you are able to show these.

*We thought that it could be relevant for the reader to know the actual elevation of the lake site. We will revise the figures (see also our reply to referee #1) and will add the explanations of the shaded areas as requested below in the caption.*

Figure 1 – what is the blue shaded bar? (also in figures 2 and 3)
*This is the area from which we calculate the deepest point. We will add explanation.*

Figures 2 and 3 – what are the red bars? (also true for Figure 7)
*This highlights those measurements that are likely connected to the occurrence of surface water. We will add explanation.*

Figure 6 – what is the grey-blue bar around the points? Need to clarify. Why do only (a) and (d) have red lines?
*This is the error bar on the volume calculation based on the scaling factor between volume and depth. Since we do not have CS2 measurements over Lake 2 and 3 it does not apply for those. We will add explanation.*

**References**

*Liang, Q., Xiao, W., Howat, I., Cheng, X., Hui, F., Chen, Z., Jiang, M., and Zheng, L.: Filling and drainage of a subglacial lake beneath the Flade Isblink ice cap, northeast Greenland, The Cryosphere Discussions, pp. 1–17, 2022.*

*Livingstone, S. J., Clark, C. D.,Woodward, J., and Kingslake, J.: Potential subglacial lake locations and meltwater drainage pathways beneath the Antarctic and Greenland ice sheets, Cryosphere, 7, 1721–1740, https://doi.org/10.5194/tc-7-1721-2013, 2013.*

*Livingstone, S. J., Li, Y., Rutishauser, A., Sanderson, R. J., Winter, K., Mikucki, J. A., Bj.rnsson, H., Bowling, J. S., Chu, W., Dow, C. F., et al.: Subglacial lakes and their changing role in a warming climate, Nature Reviews Earth & Environment, pp. 1–19, 2022.*

Chandler, D. M., Wadham, J. L., Lis, G. P., Cowton, T., Sole, A., Bartholomew, I., Telling, J., Nienow, P., Bagshaw, E. B., Mair, D., Vinen, S., and Hubbard, A.: Evolution of the subglacial drainage system beneath the Greenland Ice Sheet revealed by tracers, Nature Geoscience, 6, 195–198, https://doi.org/10.1038/ngeo1737, 2013

Palmer, S., Mcmillan, M., and Morlighem, M.: Subglacial lake drainage detected beneath the Greenland ice sheet, Nature Communications, 6, https://doi.org/10.1038/ncomms9408, 2015. Magnússon, E., Rott, H., Björnsson, H., and Pálsson, F.: The impact of jökulhlaups on basal sliding observed by SAR interferometry on Vatnajökull, Iceland, Journal of Glaciology, 53, 232–240, 2007.

Stearns, L., Smith, B., and Hamilton, G.: Increased flow speed on a large East Antarctic outlet glacier caused by subglacial floods, Nature Geosci, pp. 827—831, https://doi.org/10.1038/ngeo356, 2008 Stearns et al., (2008).

Willis, M. J., Herried, B. G., Bevis, M. G., and Bell, R. E.: Recharge of a subglacial lake by surface meltwater in northeast Greenland, Nature, 223–227, https://doi.org/10.1038/nature14116, 2015

---

## Author Response (AR1)

*Dear editor and referees,*

*We again thank the referees for their feedback on our manuscript, and suggestions for improvements. We have previously replied in detail to the review and indicated how we would propose to revise the manuscript. We have now revised the manuscript significantly with the main revisions being:*

*- Expanding and restructuring Section 3 and changed the title to "Data Seat and Data Processing". This includes a more detailed description of TanDEM-X data (Sect. 3.1) including an example of the data (Fig 2).*

*- Revised and extended description of the CryoSat-2 data (Sect 3.3). Here we*

*have included a new figure (Fig. 3) showing the waveforms from different times over Lake 1 as an example of what the CryoSat-2 waveform looks like.*

*- Included a section on "Filtering of CryoSat-2 data" (Sect. 4.1) which includes two new figures: Figure 4 shows the spatial coverage of CryoSat-2 data over Lake 4 for different processing steps and choices. Figure 5 shows processed swath points using different thresholds and the associated waveform over Lake 1, to improve the understanding of the data.*

*- We have removed the calculations and discussions of basal melt rates (l. 517-523) and Subglacial lake volumes (l. 324-344) from the manuscript.*

*- General revisions throughout the manuscript.*

*In the following, we will reply in detail again to all issues raised by the referees and explain how we have revised our manuscript accordingly.*
*We show the referee's comments in black and our response in blue italic text. Line and section numbers in our reply refer to the revised manuscript including track changes. The line/section numbers in black refer to the first version of the manuscript.*

*We hope that you will find that the revisions will have improved our manuscript.*

*On behalf of the authors,*

*Louise Sandberg Sørensen*

**Specific reply to referee #1**

This manuscript uses a combination of Cryosat-2 laser altimetry and DEMs from SAR and optical measurements to provide detailed measurements over four previously identified subglacial lakes in Greenland, and one prospective, but not previously identified lake.  It provides some details of a set of techniques for combining measurements from these sensors, and offers a longer time series of elevation changes for the lakes than previous studies did, with somewhat more temporal detail.  The use of Cryosat-2 data allows the authors to measure the depth of the lake under Flade Isblink immediately after its drainage, and finds a depth for the collapse feature that is significantly deeper than that measured in previous studies.  I had trouble identifying the scientific questions that the study answered.  Since four of the lakes had been identified in previous studies, the fact of their existence is not news, and the behavior documented in this study is not especially surprising.

*The objective of our analysis was never to document any surprising behavior of the subglacial lakes investigated. Contrarily, we wanted to investigate whether CS2 SARIn data and TanDEM-X data can be used to improve monitoring of subglacial lake activity in Greenland, and therefore, we chose those lakes that were already described in the literature as these provided the possibility to benchmark our data.*
*By documenting that these data are actually useful for monitoring the lake activity, they can/should be included in future subglacial lake studies. This has been emphasized in l. 10-12.*

The fifth, potential lake identified here is extremely small and is close to one of the previously known lakes, so I am not sure what significance I should attach to its existence.

*We agree with the referee that this lake is small, and we do not claim that it will have great importance in the overall hydrological system or in the runoff from that basin. In spite of its small size, we do think that it is important to document our findings since the active subglacial lake activity is one of the very few ways of actually observing what is happening beneath the ice sheet. We also think that the fact that two lakes might be*

*connected is interesting since this can provide some information about the hydrological pathways.*

The study may be interesting to researchers with a deep knowledge of, and interest in, the particular subglacial lakes studied here, but I am not sure how wide this audience is likely to be.

*We are sorry to learn that the referee thinks that this study will not be interesting to a larger audience. We do, however, not share that point of view. For the entire scientific community that works on subglacial lakes/hydrology, we believe that it is an important conclusion that additional datasets can be used to improve future monitoring efforts.*

The authors suggest that measurements over subglacial lakes have the potential to inform our understanding of subglacial water flow, but I really didn't see much development of this potential in this study. The abstract identifies the demonstration of techniques as a goal of the study, but the technical discussion of the techniques is brief and the presentation of the measurements is not very detailed. I would recommend reworking the study, either to focus on how each of the techniques performed at lake 4 (which had very large relief and elevation change) and at lakes 2 and 3 (which were small, and where the Cryosat-2 data didn't work well), or to try to better understand the implications of the measurements for the subglacial hydrology of the ice sheet.

*We see that referee #2 also states that it would be beneficial to rework the manuscript to make the objective clearer. We have re-structured and re-focused the manuscript to include more information on the data, including uncertainties, quality and methods.*

Line 34: Should note that this possibility was investigated in some detail by other studies (Stearns 2008, https://www.nature.com/articles/ngeo356) (Smith et al, 2017 (cited in the manuscript) And (Zwally and others, 2002, https://www.science.org/doi/10.1126/science.1072708), and that net dynamic changes after very large water inputs were negligible.

*We assume here that the referee is referring to Lines 32-34 and the statement that: "The sudden drainage and outburst flood of a subglacial lake might temporarily affect ice flow velocities downstream from the lake location Palmer et al., 2015; Liang et al., 2022)."*

*We agree with the referee here, which is also why we have written that it might impact ice velocities. We do not agree however that all the papers listed by the referee conclude that the effect is negligible. Contrarily, some quotes from those papers are:*

*"Our findings provide direct evidence that an active lake drainage system can cause large and rapid changes in glacier dynamics." (Stearns et al., 2008 )*

*"The indicated coupling between surface melting and ice-sheet flow provides a mechanism for rapid, large-scale, dynamic responses of ice sheets to climate warming." (Zwally et al, 2002).*

*The Zwally et al (2002) paper focuses on surface melt and not subglacial lakes though, so we do not see the relevance here – even though the surface and basal hydrology are connected. We agree that the Smith et al., 2017 paper describes a case where no connection between drainage and ice velocity is observed, but we do not see how this contradicts our statement in the manuscript.*

*We have revised the paragraph in the manuscript l 40-42.*

Line 88: "Classified" is not the right verb here.  "Asserted" might be better

*This sentence has been rephrased (l. 108-109).*

Section 3-1:   Is there any way the selection of thresholds can be formalized?  The thresholds selected here seem ad hoc, and it would be useful to discuss how they were chosen.

*We agree that the threshold selection seemed ad hoc. We find that the threshold is very case-specific and dependent on e.g., surface conditions (scattering properties), the geometry of the satellite orbit versus lake location, and the geometry of the surface depression. We have included figure 5, which shows the processed swath point from a single waveform and the effects of applying different threshold choices. We have also elaborated on the data processing description in Sect. 4.1.*

Line 140: "highly dynamic" should be "rough"

*This sentence is no longer relevant and has been deleted (l. 241)*

Line 141: Is the incoherent component in the processing, or in the radar reflections?

*This sentence is no longer relevant and has been deleted (l. 242)*

Line 145: should be "assumed to be representative"

*Agree. It has been revised accordingly (l. 246).*

Line 145: "were deemed as errors" should be "were assumed to be errors"

*This sentence is no longer relevant and has been deleted (l. 247)*

Line 146: "Across swath tracks close to the basin rim" should be "swath-processed data from tracks close to the basin rim" .

*This sentence is no longer relevant and has been deleted (l. 248)*

Line 148: remove commas around "which is removed"

*We have revised sentence (l. 252).*

Line 183 "vertical alignment" should be "vertical offset"

*We have revised accordingly (l. 288).*

Line 184: delete "found to be"

*We have revised accordingly (l. 288).*

Line 197: "but we see" should be "and we see"

*We have revised accordingly (l. 302).*

Line 201: "such as" -> "including"

*We have revised accordingly (l. 308).*

Line 211: What is the basin shapefile?

*We have clarified, that we refer to the manually delineated basin outline (l. 318).*

Figure 1 (and all similar figures)

*We assume that this comment is actually about figures 7-11*

1. The map extent is too broad to give a useful context for the lake location. Should instead show a context map with the regional topography and the locations of adjacent glaciers in some detail, with a reference map in a separate figure to indicate the locations of figures 1-7

*We agree that these figures can be improved. We have included a figure to show the locations of all the lake sites (Fig. 1).*

2. Need to provide a color bar for panel c

*Agree, we have revised Figures 7-11 accordingly.*

3. The yellow lines in panel b are very hard to see

*We agree that they were hard to see. We have revised the figures (7, 8, 9, 10, 11) to make the lines more clear and changed the colormap.*

4. The range of contrast in the colors in panel b does not really allow the distinction between different CS2 dates. Different symbols should be used to denote different dates.

*We have revised the figures to include different symbols with a different colormap.*

5. The legend should explain the blue shaded bar

*We have revised accordingly.*

Line 224: Subtracting the median height does not make sense, as the offset subtracted is will depend on the height distribution of the rim. It would be better to subtract a median height anomaly relative to some reference DEM. Is this what the authors mean to say?

*This is no longer relevant, since we have chosen to remove the section on Subglacial Lake Volumes from the manuscript (l. 323-343).*

Line 226: "cubing the 2sigma": What is this, and why does it give an error estimate? This needs much more detail to explain and/or justify what is done here.

*This is no longer relevant, since we have chosen to remove the section on Subglacial Lake Volumes from the manuscript (l. 323-343).*

Line 228: Need to specify which depths and volumes are used here, and need to connect these, using consistent terminology, with the depths derived from the DEMs and from CS2. Are "the depths" referenced here the depths of the deepest point from CS2?

*This is no longer relevant, since we have chosen to remove the section on Subglacial Lake Volumes from the manuscript (l. 323-343).*

Line 231 / equation 1. How does the derivation of R and V take the error bars into account? More detail is needed.

*This is no longer relevant, since we have chosen to remove the section on Subglacial Lake Volumes from the manuscript (l. 323-343).*

Line 236: It would be useful to demonstrate how R~(t) varies in time based on the available DEM data.

*This is no longer relevant, since we have chosen to remove the section on Subglacial Lake Volumes from the manuscript (l. 323-343).*

Lines 225-236: The methodology here does not seem to capture the true uncertainty in depth (and volume) estimates based on the CS2 data. When there is a large spatial variation in elevations in the DEM data, they are assessed a large error based on the slope and roughness within the relevant part of the lake, but CS2 data generally give a small number of elevation measurements at these times, and are assessed a smaller error. Would it not make sense to apply roughness information from the DEMs to the CS2 data to assess their errors?

*This is no longer relevant, since we have chosen to remove the section on Subglacial Lake Volumes from the manuscript (l. 323-343).*

Line 246: add comma after "coverage"

*We have revised accordingly (l. 352).*

Line 264: It would be useful to explore why CS2 did not provide data over lakes 2 and 3. Were there no footprints that intersected the lake boundary? Was the coherence too low?

*The lack of data is likely due to the fact that the collapse basins are small, making it difficult to differentiate multipeaked waveforms as a reflection from both a depression and a surface. Furthermore, the narrow structure of the depressions also increased the incoherent component of the phase difference, thus making it tough to do a phase unwrapping. We hope that the inclusion of Figures 3 and 5 will help the readers understanding of the challenges associated with obtaining useful data from the smallest collapse basins.*

Line 265: Please show the power image from TanDEM-X for early 2011. It would be interesting to know if there are any reflectance features associated with the about-to-drain lake.

[Figure]

*TanDEM-X SAR amplitude image of Lake2 (SouthernLakes) from 20-01-2011 (left) and optical image from Google Earth from 09/2012.*

*Interestingly, the subglacial lake and its northwestward-flowing channel appear slightly darker than their surroundings in the TanDEM-X amplitude data. There are other darker structures nearby, so identifying the subglacial drainage structures based on SAR amplitude alone does not seem sufficient. However, it could help to identify and locate them, and we have not included them in the revised manuscript.*

[Figure]

*left the southern lake (Lake2); right the DEM from 20-01-2011 TanDEM-X acquisition*

Line 279: "CS2 point data" :should this be "CS2 swath data"

*We have revised accordingly (l. 388)*

Volume calculations: Except for Flade Isblink, these volumes are exceedingly small.  Compared to lake discharges in Antarctica, they are miniscule, and those Antarctic discharges had almost no effect on ice dynamics.  What is the justification for saying that the lakes studied here might be important for ice dynamics?

*As mentioned earlier, there are references for how subglacial lake drainage can affect ice velocities.*

320: Should compare volume-change estimates against surface runoff estimates from (e.g.) RACMO.

*We agree with the referee that a study that includes both estimates of basal and surface melt with the subglacial lake activity would be interesting. This would however require modelling/observations of how much of the surface melt water that reaches the bed, which we believe is outside the scope of the current manuscript. Also, since there is a wide spread in the predicted runoff estimates from different RCMs such a study should include several models (Fettweis et al., 2020). Therefore, we have chosen to remove the subglacial lake volumes and basal melt calculations from the revised manuscript.*

358: "shortly" should be "briefly"

*We have revised the sentence (l. 477-478)*

373: "off-nadir" should be "off nadir"

*We do believe that off-nadir is the correct term here (l. 493).*

376-384: this repeats material found in the methods section.

*We agree and have deleted the sentence l. 496-500, but we have kept the last part (500-502) that emphasizes that we do not take the associated error into account.*

378: delete "parameters"

*Has been deleted.*

387: is "highly active" all that can be determined here?  This doesn't seem like a lot has been learned.

*As the focus of the revised manuscript is more on data and methods and less on the geophysical interpretations, we have deleted this sentence (507-511).*

Section 6.6

To conclude that the activity of the new potential lake affected the drainage of lake 2, the authors would need to present evidence that it is unusual for water to reach the bed in volumes comparable to those discharged by the new lake.  Looking at the images in appendix B, it appears that there is abundant water on the surface of the glacier, and it seems likely that this water often drains through moulins.  Why, then, should we believe that the drainages of lakes 2 and 3 are anything but coincidental?  Even if they were not coincidental, what specifically does this tell us about the hydrology of the glacier bed that we could not have inferred already?

*This is true. Here, we simply want to point to the fact that the timing of the events could imply that they are connected. We do not foresee to do any detailed analysis in this work to support this hypothesis. But we agree that the section needed to be improved by expanding on the information and discussion, and we have done that (Section 6.5).*

Appendix A:  Why would the basal melt rates be important in this area?  Water fluxes from surface melt must dwarf these rates by orders of magnitude.  Please consider surface melt first.

*We have removed the basal melt plot and associated discussion from the manuscript.*

Appendix B:

Figure B1:  Indicate the location of this lake relative to lake 2.

*We have done this in Fig A2*

Also- what is being mapped here?  The difference between panels a and b seems to mostly be that in panel B the surface is covered with snow, while in panel A it is mostly bare ice.  The interpretation of the change in the collapse basin is not at all clear to me.

Figure B2: There is a lot of variability in surface conditions between these images.  The interpretation in the text is not at all convincing.

*We agree and have revised the Appendice figures and the discussion of it in Sect 6.5*

Data availability: I didn't see a statement about data availability for the CS2 swath-mode data.

*We will be happy to make the data available. We will do so on data.dtu.dk and provide the link if the manuscript is accepted.*

*References*

*Fettweis, Xavier, et al. "GrSMBMIP: intercomparison of the modelled 1980–2012 surface mass balance over the Greenland Ice Sheet." The Cryosphere 14.11 (2020): 3935-3958.*

*Palmer, S., Mcmillan, M., and Morlighem, M.: Subglacial lake drainage detected beneath the Greenland ice sheet, Nature Communications, 6, https://doi.org/10.1038/ncomms9408, 2015.*

*Liang, Q., Xiao, W., Howat, I., Cheng, X., Hui, F., Chen, Z., Jiang, M., and Zheng, L.: Filling and drainage of a subglacial lake beneath the Flade Isblink ice cap, northeast Greenland, The Cryosphere Discussions, pp. 1–17, 2022.*

*Stearns, L., Smith, B., and Hamilton, G.: Increased flow speed on a large East Antarctic outlet glacier caused by subglacial floods, Nature Geosci, pp. 827––831, https://doi.org/10.1038/ngeo356, 2008 Stearns et al., (2008).*

*Smith, B. E., Gourmelen, N., Huth, A., and Joughin, I.: Connected subglacial lake drainage beneath Thwaites Glacier,West Antarcticaf, The Cryosphere, 11, 451–467, 2017.*

*Zwally, H. J., Abdalati, W., Herring, T., Larson, K., Saba, J., & Steffen, K. (2002). Surface melt-induced acceleration of Greenland ice-sheet flow. Science, 297(5579), 218-222.*

**Specific reply to referee #2**

**General Comments**

This paper combines multiple satellite missions to improve the temporal resolution of ice surface elevation change measurements over 4 previously identified active subglacial lakes in Greenland to provide new constraints on lake volume and evolution. In addition, they find one potential new active lake that might be hydrologically connected to one of the known lakes (although see specific comments). The study is generally well written with some nice figures, and I found the combination of methods to improve the temporal resolution convincing. I did, however, find quite a few minor errors or places which needed further clarification

(see specific comments below), and I agree with the other reviewer that the implications of their findings are currently not clear, and could do with expanding / reworking. For example, could you combine your improved monitoring of recharge rates with your basal melt modelling (expanded to all sites), to make this a more significant component to better explore the role of surface vs basal melt. How do your recharge rates/ drainage rates compare to elsewhere? Can you use your improved timings of drainage to better link to triggers?

*As also mentioned in our reply to referee #1 the aim of our study has been to investigate whether CS2 SARIn data and TanDEM-X data can be used to improve monitoring of subglacial lake activity in Greenland. Both referees suggest that the manuscript is reworked to make its aim clearer. We have revised the manuscript to focus on the data and its usefulness in subglacial lake monitoring. We have removed the calculations and discussions on basal melt and subglacial lake volumes from the revised manuscript, and instead improved and expanded the data method section and associated discussion.*

**Specific Comments**

L4 – Antarctic Ice Sheet

*We have revised accordingly.*

L6 – I think it would be worth mentioning earlier in the abstract that active lakes are typically identified from ice-surface elevation changes to put this point into context.

*Agree. We have added the following sentence to the abstract: "Active lakes may be identified by local changes in ice topography caused by drainage or recharge of the lake beneath the ice." (l. 5-6)*

L14 – It is odd to mention surface hydrology at the end as this is not discussed in the rest of the abstract.

*Agree. We have deleted this sentence from the manuscript (l. 18-19)*

L21 – not sure this reference is appropriate here as it focuses on predicting lake locations. Perhaps refer to the Livingstone et al. (2022) study instead.

*Agree. This was a mistake. We are now referring to Livingstone et al. (2022) and Fan et al., 2023 instead of Livingstone et al. (2013). L. 26.*

L24 – "steeper ice surface slopes"

*We have revised accordingly. L 30.*

L27 – delete "further". Your previous points were around different settings not detection.

*We have revised accordingly. L. 33.*

L30 – the use of e.g. in this sentence does not work that well. Can you combine the first part of this sentence with the second part of the next to provide a more general mechanism for lake drainage?

*We have revised the sentence to:*

*"The lake will eventually drain when filled with enough water to resist the pressure exerted by the overlying glacial load (Chandler et al., 2013), hence a subglacial lake drainage events can be triggered by a prolonged addition of surface meltwater (Livingstone et al., 2022)." L. 36-38.*

L33 – I think Palmer look at vertical displacement, but don't really mention horizontal displacement. It might be better to refer to some of the key velocity studies in Antarctica or Iceland here.

*We have changed the reference from Palmer et al., (2015) to Magnusson et al, (2007) and Stearns et al., (2008). L. 40.*

L35 – suggest – "Notably, the period of lake recharge following a drainage event provides..."

*We have changed this sentence to: "In particular, the period of lake recharge provides information about subglacial water production and conditions at the bed" L. 44.*

L49 – should this be InSAR not SARIn?

*We made an error here. We have rephrased to: "synthetic aperture radar interferometric (SARIn) mode mode". L. 64-65.*

L56 – delete "source of" to avoid repetition of this word.

*We have revised accordingly. L. 72.*

L59 -this makes it sound like there have just been 4 active lakes identified in Greenland. There are actually 7 – see Livingstone et al. (2019). Worth noting this and rephrasing to justify the four you have chosen.

*We have rephrased the sentence (l. 83-84). We have also added the sentence "We have chosen not to include the three subglacial lakes located beneath the highly dynamic Isunnguata Sermia glacier due to the small size of the collapse basins and their location very close to the ice sheet margin (Livingstone et al. 2019)." . L. 87-89.*

L66 – this makes it sound like Bowling et al. (2019) identified all these lakes. It would be better to cite the original papers for all these lakes.

*We have removed this statement as the relevant papers are cited in each subsection about the specific lakes. L. 83.*

L72 – Figure 2a is cited before Figure 1.

*We will include a new figure 1 int the revised manuscript, which shows the location of all of the lakes and this is the first figure we refer to. (P. 4)*

L83 – is this a subsidence event in 2004? Not clear.

*Howat et al, 2015 states that "Drainage occurred in two episodes: a smaller event in 2004 and a larger one in 2011."*

*We have revised the sentence (l. 97-99) to: "These studies find that Lake 1 has drained both in 2004 (smaller event) and one in 2011 (larger). The 2011 drainage occurred with an unknown rate within two weeks (28 June, 2011 to 12 July, 2011), resulting in the formation of a collapse basin in the ice sheet surface."*

L94 – suggest: "… which could indicate some recharge of the subglacial lakes by surface water".

*We have revised accordingly. L.117.*

L97 – Flade Isblink Ice Cap

*We have revised accordingly. L.119.*

L105 – clarify whether this was infill of the collapse surface basin or infill of the subglacial lake by surface water causing the ice surface to rise.

*We have revised the sentence to:*

*"The elevation of the collapse basin rapidly increased by 30 meters over the following two years due to inflow of surface water to the subglacial lake, and between August 2012 and April 2013 a topographic bulge appeared in the basin (Willis et al., 2015)." L. 127-129.*

L107 – can you quantify the elevation change associated with this event?

*Liang et al (2022) observed an elevation change of 10 m. We have added this to the text. L. 129-130.*

Section 3 – it would be useful to provide details on the final resolution and vertical/horizontal errors of the processed datasets.

*We have have revised and description of both the TanDEM-X (Sect 3.1) and CryoSat-2 (Sect. 3.3 and 4.1) data in the revised manuscript to hopefully provide the reader with a better understanding of the datasets. This includes addition of four new figures (Fig. 2, 3, 4, 5) of the data itself and the impact of the different processing steps for the CryoSat-2 swath data.*

L112 – SARIn has already been defined.

*We have revised accordingly. L. 163*

L124 – It would be helpful to quantify this change in density.

*The density of the swath-processed data compared to the POCA varies and depends on e.g. the thresholds used in the swath processor, and the physical properties and topography of the surface, which affects the waveform. We will add the following sentence to the paragraph (L. 176-178):*

*"Depending on e.g., the chosen processing thresholds, the physical properties and the topography of the area, the L2 swath processing leads to a 10 to 100 folds increase in elevation measurement compared to conventional L2 processing."*

*We have included a figure to show the data coverage of POCA and different processing/filtering steps of the CryoSat-2 swath data. (Fig. 4).*

L128 – "… thresholds compared to those usually applied …."

*We have revised accordingly (L. 191).*

L134-138 – Maybe this is because I am a visual person, but a figure showing the raw to processed data points would be really helpful here in allowing the reader to judge the effectiveness of the approach.

*Have included a figure (Fig. 5) which shows an example of a CryoSat-2 waveform at Lake 1 together with the derived elevation estimates and color/symbol coded these to show their associated coherence and range bin number.*

L143 – not sure why you need the word "apparent"?

*This sentence has been revised (L. 240-247)*

L145 – "assumed to be representative"

*We have revised accordingly. L. 246*

L146 – how close in time? This is rather vague and could do with rephrasing.

*This sentence is not longer relevant and has been deleted. (L. 247)*

148 – would be useful to get a rough estimate – 1%, 10%, 90%? See my comment above, a figure showing the stages in the processing would be helpful here.

*We have included a figure (Fig. 4) to illustrate the data coverage and how it is affected by different processing choices and on the filtering. We have further explained in l. 250-253 that it also varies with physical setting and properties of each site.*

L160 – "data takes located"? Is there an error here, I didn't follow this part of the sentence?

*The TanDEM section (Sect. 3.1) has been revised (l. 144)*

L181 – "the vertical bias."

*We have revised accordingly. L. 285.*

L192 – given you correct using local ice flow, it would be useful to know here whether this 100 m in 10 years equates to a local ice flow in this region of ~10 m/yr.

*We are not sure what the referee is asking for here.*

L198 – "One reason for this is that the…" I don't really get the point around the size of the lake as surely it is the lake edge that you are tracking.

*Correct. We have revised the sentence to: "One reason for this can be that the subglacial lake drains again in the observational period, which could make the potential ice flow less evident since the collapse basin is re-formed over the stationary location of the subglacial lake." L. 303-305.*

Section 4.2 – are these calculations still based on the local difference compared to the basin rim? If not, could these not be influenced by the different penetration depths etc? I don't really follow the approach to calculating the deepest depths (why take a mean if looking for deepest point) or the use of standard deviation. Is this not just a measure of roughness of the floor of the basin? This needs clarifying.

*These calculations are the absolute elevations of the aligned datasets. Since the datasets all are aligned at the rim (section 4.1) they will not be affected by different penetration depths. L. 280-284*

*We agree that the mean and standard deviation is also a measure of the roughness of the floor of the basin, but these will inherently have an impact on the accuracy of the measurements.*

L226 – In other papers error is calculated by multiplying the internal error of the DEM by the lake area both before and after drainage and adding together in

quadrature. What is the basis for your approach, especially as you state that 2 standard deviations is not a true measure of their accuracy?

*In the revised manuscript we have deleted the section on Subglacial Lake Volumes.*

L260 – does this actually show uplift? Could you run regression analysis over the two periods to calculate the recharge rates more accurately.

*No, not a significant uplift. We will revise the sentence to:*

*"The subglacial lake recharge can be divided into a fast basin uplift of ~13 m/yr in the period 2011-2015, and a period with no significant uplift from 2015-2019." L. 367-369.*

L276 – Flade Isblink Ice Cap

*We have revised accordingly. L. 385.*

Section 5.1 – Some of the text in this section would I think be better incorporated into the figure captions e.g., "To maintain a visually clear plot not all data sets are shown in Figure 5b." This would help the flow of this section while making the figures standalone.

*We have chosen to keep the specific sentence in the section 5 but have made some further revisions of the text. We have revised the figure captions in this section (Fig 8-11),*

L297 – Although this is the maximum volume measured, it is the minimum possible volume (given you might have missed the period of maximum collapse (i.e. it might have collapsed and then recharged between data points).

*We have chosen to delete the section on Subglacial Lake Volumes in the revised manuscript.*

L310 – can you quantify this – re. number of data points over X years?

*This is very site specific.*

L313-315 – There does seem to be some signal of the final collapse and then beginning of the recharge period though during the winter 2011/2012 period. To better test this it would be better to split these components and calculate the recharge for the upwards tick as a rate vs. the summer after.

*What we meant was that we do not see that there is a recharge in late 2011 when taking error bars into account. We will rephrase to make it more clear: "Notably, the addition of CS2 observations during 2011/2012 allows us to conclude that no significant recharge of the subglacial lake occurred in the second half of 2011, while recharge is observed throughout 2012." 427-429.*

L316 – How do you know it is bedrock? Suggest change to "bed"

*We have revised accordingly. L. 431.*

L318 – Ok, but could this not be associated with a decrease in filling over time as the area increases (i.e. for a given melt input the rate will decreases because of the basin shape?). I think it is fairly common for recharge rates to slow over time.

*Good point. We have rephrased the sentences to: "We further hypothesize that the infilling of the collapse basin after 2014/2015 is likely primarily caused by snowfall and ice flow, and less recharging of the subglacial lake due to the fact that the center of the collapse basin moves away from the subglacial lake as a result of local ice flow. The filling rate will however usually slow over time due to the geometry of the lake." L. 431-437.*

L320 – "model estimates of basal melt rates" – ok, based on what data? Need a supporting reference and to quantify. I don't quite see how this point fits with the idea of ice flow/ snowfall. What is the surface mass balance change?

*We have removed the basal melt estimates and discussion from the manuscript as part of shifting the focus towards the data and methods.*

L321 – Appendix not Append.

*This paragraph is no longer relevant and has been deleted (L. 436)*

L322 – capitalise vatnajokull ice cap

*We have revised accordingly. L. 438.*

L330 – A more positive spin would be to give the time span over which the drainage could have happened.

*We have chosen to keep the original wording here.*

L331 – "drains"

*Since we are here discussing two lakes we believe that it should be "drain" and not "drains"? L. 447.*

L332 – "spring"

*We have revised accordingly. L. 448*

L334 – In which case, how do you know whether Lake 2 is actually a supraglacial lake that is filling and draining? It would be useful to confirm whether the 2011 data is associated with surface water or not.

*Figure 8(b) shows that the surface over Lake 2 was not flat in 2011, which we assume would be the case if it was a supraglacial lake. We have indicated with light red boxes in Figures 8(d) and 9(d) times where we identified surface water in optical imagery. This explanation has been added to the caption text of these figures.*

L347 – please state the infilling rate from this calculation.

*We have stated the infilling rate (L. 463-464)*

L352 – It would be useful to incorporate the results of Liang et al. (2022) into this discussion as they look at seasonal recharge (i.e. impact of warmer summers on recharge rate, with rates of up to 49 m/yr).

*We have added a paragraph about the Liang et al. (2022) results in Sect. 6.3. L. 471-479.*

L357-358 – "which shortly affected the local horizontal ice velocity" – needs rephrasing.

*We have rephrased the sentence to: "The surface lowering in 2019 is also documented from ICESat-2 data by Liang et al. (2022), who also identified it as a drainage event. They also document that this event caused the ice velocity downstream from the lake to abruptly but shortly increase. " L. 476-478.*

L368 – "basins are flat"

*We have revised accordingly. L. 488.*

L378 – You use SARIn elsewhere.

*Yes, but this is correct. For CryoSat-2 the data is called SARIn.*

L386 – Ok, but others have shown this in their data, so should acknowledge this.

*We have deleted this sentence (L. 507-511).*

L387-388 – this needs clarifying? Where is this low pressure region?

*We have deleted this sentence (L. 507-511).*

Section 6.5 – this seems rather tagged on and is not explained in the methods, and very briefly here. I would suggest introducing earlier in the method section and showing in the results, or deleting.

*We agree and the Section on Basal Melt Flux is not part of the revised manuscript.*

L402 – There is also a new paper in TCD by Fan et al. (2022) with lots of new active subglacial lakes. Is this one of the lakes in their inventory?
https://tc.copernicus.org/preprints/tc-2022-122/

*We have included the Fan et al. (2023) as reference. (L. 26, 59-60, 513-515, 552-554).*

Section 6.6 – given that this period coincided with a large melt event could these both be responding independently to large volumes of water accessing the bed in this region? More information is needed here to better test this hypothesis.

*This is true. Here, we simply want to point to the fact that the timing of the events could imply that they are connected. We do not foresee to do any detailed analysis in this work to support this hypothesis. But we have revised the Section (Sect 6.5) to better support it.*

L422 – Greenland Ice Sheet.

*We have revised accordingly.*

L423 – "investigated elevation changes"
*We have revised accordingly.*

Figures/ Tables

Given that you correct based on the elevation outside of the collapse basins, I was surprised to see many of the plots with a y-axis of elevation rather than elevation anomaly (relative to the tie points). It would be useful to clarify whether you use similar time periods as cross-over points or how you are able to show these.

*Our reasoning for using the absolute elevations rather that relative ones is it could be relevant for the reader to know the actual elevation of the lake site.*

Figure 1 – what is the blue shaded bar? (also in figures 2 and 3)

*We have revised the figures have added colour bars (Figs. 7(c), 8(c), 9(c), 10(c), 11(c)).*

Figures 2 and 3 – what are the red bars? (also true for Figure 7)

*This highlights those measurements that are likely connected to the occurrence of surface water. We have added the explanation in the figure captions. (Figs. 8(d), 9(d), 11(d)).*

Figure 6 – what is the grey-blue bar around the points? Need to clarify. Why do only (a) and (d) have red lines?

*This figure is not included in the revised manuscript.*

**References**

*Liang, Q., Xiao, W., Howat, I., Cheng, X., Hui, F., Chen, Z., Jiang, M., and Zheng, L.: Filling and drainage of a subglacial lake beneath the Flade Isblink ice cap, northeast Greenland, The Cryosphere Discussions, pp. 1–17, 2022.*

*Livingstone, S. J., Clark, C. D.,Woodward, J., and Kingslake, J.: Potential subglacial lake locations and meltwater drainage pathways beneath the Antarctic and Greenland ice sheets, Cryosphere, 7, 1721–1740, https://doi.org/10.5194/tc-7-1721-2013, 2013.*

*Livingstone, S. J., Li, Y., Rutishauser, A., Sanderson, R. J., Winter, K., Mikucki, J. A., Bj.rnsson, H., Bowling, J. S., Chu, W., Dow, C. F., et al.: Subglacial lakes and their changing role in a warming climate, Nature Reviews Earth & Environment, pp. 1–19, 2022.*

*Livingstone, S. J., Sole, A. J., Storrar, R. D., Harrison, D., Ross, N., & Bowling, J. (2019). Brief communication: Subglacial lake drainage beneath Isunguata Sermia, West Greenland: geomorphic and ice dynamic effects. The Cryosphere, 13(10), 2789-2796.*

*Chandler, D. M., Wadham, J. L., Lis, G. P., Cowton, T., Sole, A., Bartholomew, I., Telling, J., Nienow, P., Bagshaw, E. B., Mair, D., Vinen, S., and Hubbard, A.: Evolution of the subglacial drainage system beneath the Greenland Ice Sheet revealed by tracers, Nature Geoscience, 6, 195–198, https://doi.org/10.1038/ngeo1737, 2013*

*Palmer, S., Mcmillan, M., and Morlighem, M.: Subglacial lake drainage detected beneath the Greenland ice sheet, Nature Communications, 6, https://doi.org/10.1038/ncomms9408, 2015.*

*Magnússon, E., Rott, H., Björnsson, H., and Pálsson, F.: The impact of jökulhlaups on basal sliding observed by SAR interferometry on Vatnajökull, Iceland, Journal of Glaciology, 53, 232–240, 2007.*

*Stearns, L., Smith, B., and Hamilton, G.: Increased flow speed on a large East Antarctic outlet glacier caused by subglacial floods, Nature Geosci, pp. 827––831, https://doi.org/10.1038/ngeo356, 2008 Stearns et al., (2008).*

*Willis, M. J., Herried, B. G., Bevis, M. G., and Bell, R. E.: Recharge of a subglacial lake by surface meltwater in northeast Greenland, Nature, 223–227, https://doi.org/10.1038/nature14116, 2015*

*Fan, Y., Ke, C.-Q., Shen, X., Xiao, Y., Livingstone, S. J., and Sole, A. J.: Subglacial lake activity beneath the ablation zone of the Greenland Ice Sheet, The Cryosphere, 17, 1775–1786, https://doi.org/10.5194/tc-17-1775-2023, 2023.*

---

## Author Response (AR2)

*We would like to thank the two referees for their evaluation of our revised manuscript. We are happy to see that they found that our manuscript has improved from our revisions. Below are our replies to their comments. The referees' comments are in* black text *and our replies are in blue italic.*

**Referee #1**

Second review of TC-2022-263

This is reviewer 1 again.

The revisions to the manuscript have improved the figures, and shifted the focus of the study in towards a description of the methods and processing strategies.

I'd like to reiterate one major point that I brought up in my first review that the authors dismissed in their response: the supposedly linked drainage of lakes 2 and 3.
*The possible linked drainage is between lakes 2 and 5.*
In my first review, I suggested that the evidence for a causal connection between the drainage of the two lakes was lacking, and suggested that the authors consider the amounts of water available from supraglacial hydrology. What I meant by this is that the amount of water coming from lake 2 is very small compared to the amount of water generated by supraglacial melt in this part of the ice sheet. If there is a connection between the drainages of lakes 2 and 3, it is much more likely to be because both drainages were triggered by an input of meltwater from the surface.
*We agree that the amount of water at the surface is much larger than what is available from the drainage of Lake 2, but this applies each summer and we only see drainages in 2012.*
The authors rejected my suggestion to use surface climatology to estimate the amount of meltwater available, but I would suggest that it is very unlikely that such a consideration would show that the water available from the lake 2 is at all significant compared to the amount of water produced at the surface. In Antarctica, when two adjacent lakes experience some change close in time to one another, the hypothesis of a causal connection is easier to argue, because there is no other source of water apart from the lakes; in Greenland, the ice is melting rapidly at the surface (especially in July and August) and water is running all over the place. The authors' conclusion that their study provides "indication of hydrologically connected subglacial lakes in Greenland" seems at best an overstatement of the evidence available. If they wish to make this statement, they should mention the likely possibility that both lake drainages were triggered by surface melt.
*We agree and have revised the last paragraph in the abstract to emphasize this. We have also changed title of section 6.5 to "New Subglacial Lake", and added the following sentence to this section: "It is also possible that the coincident drainage of the two lakes was independently triggered by the large amount of surface meltwater available in the summer of 2012."*
In the same vein, the authors' suggestion that lake drainage intervals provide information about water production at the bed (~line 35) does not seem to make a lot of sense in Greenland (though it is possibly true in parts of Antarctica).
*We do see over e.g. Lake 4 (Flade Isblink) that the lake recharges during winter as well as summer, and this is an indication of subglacial melt. We agree that it can also be an indication of transfer from the surface to the bed and we have revised this sentence accordingly.*
The new material describing Cryosat-2 processing is interesting, but not everything about the new section is clear to me. In figure 5, the swath processing has produced multiple elevation estimates at the same longitude, with xes , diamonds, and circles at the same longitude. I think this is most likely a result of an unwrapping error in the phase processing, and the 'x' measurements should be located far off to the left relative to the other measurements. The figure would be easier to interpret if the authors indicated the look direction of the waveform, plotted the x axis in meters rather than in longitude, and provided a value for the horizontal separation corresponding to a wrapped phase ambiguity.
*We agree that it is easier to read the figure with m instead of longitude on the x-axis but it does not change the figure much. We have revised the figure accordingly. We are not sure what is meant by the look angle of the waveform.*
*The reason that some swath points are geolocated to the same longitude (waveform distance) is indeed because of noise and errors (e.g. unwrapping), because we have lowered the thresholds to get as much output data as possible. This is also why a filtering is applied afterwards.*

They might also comment on the fact that the brightest part of the return is located at the bottom of the deepest surface depression, which must indicate something about how this reflection was produced.
*We have added this sentence to section 4.1: "The large peak in the waveform might be caused by the reflection from surface water at the bottom of the collapse basin."*

**Referee #2**

Thank you for carefully considering the comments from myself and the other reviewer. I think you have done a good job of addressing these, including changing the focus away from lake volumes and expanding on the methods. I have just a few, v. minor final comments, below:

Note line numbers refers to the track-changed document.
L15 – rather than "shows signs" could you be more specific here, e.g. both drained within 1 month with the upstream lake draining first, suggesting that …
*We have revised the last paragraphs of the abstract to "We also present evidence of a new active subglacial lake in Southwest Greenland, which is located close to an already-known lake. Both lakes probably drained within one month in the summer of 2012, which suggests that they are either hydrologically connected or that the drainages were independently triggered by extensive surface melt. This is to our knowledge the first indication of hydrologically connected subglacial lakes in Greenland."*
L30 – "characterized by a steeper ice surface slope"
*Revised accordingly*
L37 – overcome rather than resist (sorry, I missed this initially)
*Revised accordingly*
L50 – not strictly true as large lakes can be identified from flat ice-surfaces (e.g., Vostok). You note this below so I would end this sentence at " …sounding."
*Revised accordingly*
L86 – I would recommend removing this sentence, putting "(Fig. 1)" at the end of the previous sentence and adding the background context to the figure caption.
*Revised accordingly*
L88 – This doesn't quite make sense as the lakes under Isunguata Sermia are of a similar size to three of the four lakes you investigate (~1 km). I would add to the figure of active subglacial lakes identified from collapse basins at L60, and remove this sentence here, or shorten to focus on the fact they are close to the margin. It is fine to concentrate on a subset of lakes.
*We have revised the sentence to:*
*"We do not include the three known subglacial lakes located beneath the Isunnguata Sermia glacier in this study, due to their location in the highly dynamic region very close to the ice sheet margin (Livingstone et al., 2019)".*
L176 – "Depending on, for example, the …"
*Revised accordingly*
L232 – "when" rather than "if"
*Revised accordingly*
L221-239 – there is a lot of description of the figure here that you could consider moving to the caption, with the third paragraph of the three being the relevant text that should remain in the main text.
*The following text was moved to the caption of Fig. 5(a): "The dark green sections of the waveform are those accepted at a coherence threshold of 0.8, light blue sections are those additionally accepted from lowering the threshold to 0.7. Likewise, the yellow and pink are those for coherence thresholds of 0.6 and 0.5, respectively." and "The round points are from the early bin range at ~300, the diamond shapes are from the range at ~500, the triangles are from the large peak at ~600, and the crossed points are from the low power section after the large peak."*
L431 – I understand what you mean, but can you be more explicit in linking your new elevation observations to surface melt rates in the second half of 2011 vs summer 2012.
*We have revised this paragraph to: "Notably, the addition of CS2 observations during 2011/2012 allows us to conclude that no substantial recharge of the subglacial lake occurred between August 2011 and May 2012 (see Fig. 7(b)), while recharge is observed between May and November 2012. The fact that the rate of recharge was insignificant during a period outside the melt season supports the hypothesis proposed by Palmer et al. (2015)*

*and Howat et al., (2015) that the subglacial lake is primarily driven by surface meltwater drained to the bed through moulins during the melt season."*

L432 – doesn't ice flow encompass ice deformation?

*Revised accordingly*

L488 – "Lakes 1 and 4"

*Revised accordingly*

L507-515 – It is odd to have 2 x 1 sentence paragraphs. Could these be combined or included elsewhere.

*We have moved this sentence to the end of the introduction:*

*"We do not include ICESat-2 data satellite laser altimetry in this study as the main goal has been to densify the time series covered by the CS2 mission, but we acknowledge that this sensor provides an obvious dataset for future monitoring of subglacial lake activity (Fan et al., 2023)".*

*We have deleted the sentence: "The steep and deep basins could lead to phase unwrapping errors in side-looking InSAR. In this study, we checked the InSAR elevations individually to avoid the use of phase unwrapping error distortioned DEMs."*

L530 – Can you quantify the scale of this elevation change?

*We have revised the sentence to: " Between July 18, 2012, and August 12, 2012, a rapid ~15 m surface elevation lowering occurs and a feature resembling a collapse basin is formed."*

L539-544 – This sentence is very long and not easy to read. Please consider rephrasing.

*We have revised the sentence to: " For Lake 2 we know from a TanDEM-X DEM that the collapse hasn't occurred in January 2011 (Fig. 8(b)), and Landsat-7 imagery further does not show a collapse basin either in early June, 2012 (Fig. A3). Although the image is not very clear, possibly due to snow cover, we see a body of surface water that partially intersects with the outline of the collapse basin as observed in an August ArcticDEM from 2012, indicating that a local depression had not yet formed in June 2012."*

L547 – This should be Lake 5 not Lake 2.

*Revised accordingly*

L566 – could you give an example of how much additional data these are?

*We have included the sentence:*

*"For example, the number of measurement epochs increased from five from ArcticDEM alone to 22 when including Tandem-X and CS2 over Lake 1, for the time period 2011-2018."*

L569 – Here and above, could you give a constraint (i.e. drained between X and Y) rather than single date.

*We have revised this sentence to: "Previous literature did not conclude on the timing of the drainage event over Lake 2, but the inclusion of TanDEM-X scenes shows that the drainage happened in the period between 20 January 2011 and 18 July 2012."*

Figure 1 – would be improved from having an ice sheet margin outline to show where the ice is.

*We would like to keep the figure as it is, because we think that it is interesting to see the underlying bedrock topography and adding an ice sheet margin would conflict with this. Also, the ice extent is shown Figures 7(a)-11(a) and in Fig. A2(b)*

Figure 5 – label subpanels – a, b

*We have revised the figure.*

---

## Author Response (AR3)

Dear Etienne,

Thank you for your suggestions on how to revise our manuscript. I hope that I have now phrased it in an acceptable way so that it is clear that a hydrological connection between the two lakes is not confirmed in our work, but that the timing of their drainages could possibly indicate that such a connection exists.

Also a huge thank you for your quick and thorough feedback during the entire process.

Best regards,
Louise